# Extremely Simple Activation Shaping for Out-of-Distribution Detection

**Andrija Djurisic**[1]    **Nebojsa Bozanic**[1,2]    **Arjun Ashok**[1]    **Rosanne Liu**[1,3]

## ABSTRACT

The separation between training and deployment of machine learning models implies that not all scenarios encountered in deployment can be anticipated during training, and therefore relying solely on advancements in training has its limits. Out-of-distribution (OOD) detection is an important area that stress-tests a model's ability to handle unseen situations: *Do models know when they don't know?* Existing OOD detection methods either incur extra training steps, additional data or make nontrivial modifications to the trained network. In contrast, in this work, we propose an extremely simple, post-hoc, on-the-fly activation shaping method, **ASH**, where a large portion (e.g. 90%) of a sample's activation at a late layer is removed, and the rest (e.g. 10%) simplified or lightly adjusted. The shaping is applied at inference time, and does not require any statistics calculated from training data. Experiments show that such a simple treatment enhances in-distribution and out-of-distribution distinction so as to allow state-of-the-art OOD detection on ImageNet, and does not noticeably deteriorate the in-distribution accuracy. Video, animation and code can be found at: `https://andrijazz.github.io/ash`.

## 1 INTRODUCTION

Machine learning works by iteration. We develop better and better training techniques (validated in a closed-loop validation setting) and once a model is trained, we observe problems, shortcomings, pitfalls and misalignment in deployment, which drive us to go back to modify or refine the training process. However, as we enter an era of large models, recent progress is driven heavily by the advancement of scaling, seen on all fronts including the size of models, data, physical hardware as well as team of researchers and engineers (Kaplan et al., 2020; Brown et al., 2020; Ramesh et al., 2022; Saharia et al., 2022; Yu et al., 2022; Zhang et al., 2022). As a result, it is getting more difficult to conduct multiple iterations of the usual train-deployment loop; for that reason *post hoc* methods that improve model capability *without* the need to modify training are greatly preferred. Methods like zero-shot learning (Radford et al., 2021), plug-and-play controlling (Dathathri et al., 2020), as well as feature post processing (Guo et al., 2017) leverage post-hoc operations to make general and flexible pretrained models more adaptive to downstream applications.

The out-of-distribution (OOD) generalization failure is one of such pitfalls often observed in deployment. The central question around OOD detection is "*Do models know when they don't know?*" Ideally, neural networks (NNs) after sufficient training should produce low confidence or high uncertainty measures for data outside of the training distribution. However, that's not always the case (Szegedy et al., 2013; Moosavi-Dezfooli et al., 2017; Hendrycks & Gimpel, 2017; Nguyen et al., 2015; Amodei et al., 2016). Differentiating OOD from in-distribution (ID) samples proves to be a much harder task than expected. Many attribute the failure of OOD detection to NNs being poorly calibrated, which has led to an impressive line of work improving calibration measures (Guo et al., 2017; Lakshminarayanan et al., 2017; Minderer et al., 2021). With all these efforts OOD detection has progressed vastly, however there's still room to establish a Pareto frontier that offers the best OOD detection and ID accuracy tradeoff: ideally, an OOD detection pipeline should not deteriorate ID task performance, nor should it require a cumbersome parallel setup that handles the ID task and OOD detection separately.

---

[1]ML Collective. [2]Faculty of Technical Sciences, University of Novi Sad. [3]Google Research, Brain Team. Correspondence to `andrija@mlcollective.org`.

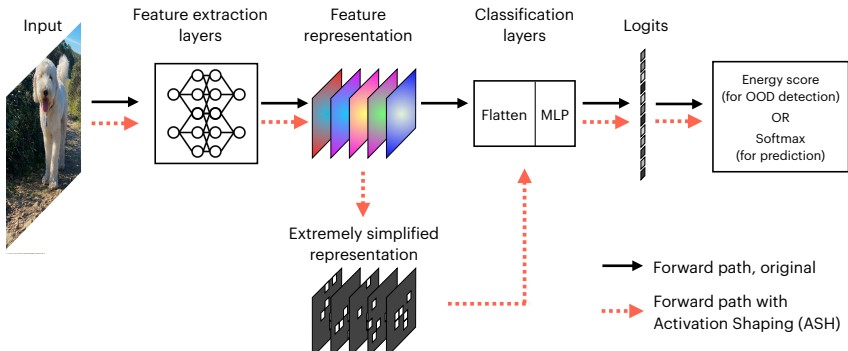

Figure 1: **Overview of the Activation Shaping (ASH) method.** ASH is applied to the forward path of an input sample. Black arrows indicate the regular forward path. Red dashed arrows indicate our proposed ASH path, adding one additional step to remove a large portion of the feature representation and simplify or lightly adjust the remaining, before routing back to the rest of the network. Note: we default to using the energy score calculated from logits for OOD detection, but the softmax score can also be used for OOD, and we have tested that in our ablation study.

A recent work, ReAct (Sun et al., 2021), observed that the unit activation patterns of a particular (penultimate) layer show significant difference between ID and OOD data, and hence proposed to rectify the activations at an upper limit—in other words, clipping the layer output at an upper bound drastically improves the separation of ID and OOD data. A separate work, DICE (Sun & Li, 2022), employs weight sparsification on a certain layer, and when combined with ReAct, achieves state-of-the-art on OOD detection on a number of benchmarks. Similarly, in this paper, we tackle OOD detection by making slight modifications to a pretrained network, assuming no knowledge of training or test data distributions. We show that an unexpectedly effective, new state-of-the-art OOD detection can be achieved by a post hoc, one-shot *simplification* applied to input representations.

The extremely simple **A**ctivation **SH**aping (**ASH**) method takes an input's feature representation (usually from a late layer) and perform a two-stage operation: 1) remove a large portion (e.g. 90%) of the activations based on a simple top-K criterion, and 2) adjust the remaining (e.g. 10%) activation values by scaling them up, or simply assigning them a constant value. The resulting, simplified representation is then populated throughout the rest of the network, generating scores for classification and OOD detection as usual. Figure 1 illustrates this process.

ASH is similar to ReAct (Sun et al., 2021) in its post-training, one-shot manner taken in the activation space in the middle of a network, and in its usage of the energy score for OOD detection. And similar to DICE (Sun & Li, 2022), ASH performs a sparsification operation. However, we offer a number of advantages compared to ReAct: no global thresholds calculated from training data, and therefore completely *post hoc*; more flexible in terms of layer placement; better OOD detection performances across the board; better accuracy preservation on ID data, and hence establishing a much better Pareto frontier. As to DICE, we make no modification of the trained network whatsoever, and only operate in the activation space (more differences between ASH and DICE are highlighted in Section K in Appendix). Additionally, our method is plug-and-play, and can be combined with other existing methods, including ReAct (results shown in Table 5).

In the rest of the paper we develop and evaluate ASH via the following contributions:

- We propose an extremely simple, post-hoc and one-shot activation reshaping method, ASH, as a unified framework for both the original task and OOD detection (Figure 1).

- When evaluated across a suite of vision tasks including 3 ID datasets and 10 OOD datasets (Table 1), ASH immediately improves OOD detection performances across the board, establishing a new state of the art (SOTA), meanwhile providing the optimal ID-OOD trade-off, supplying a new Pareto frontier (Figure 2).

- We present extensive ablation studies on different design choices, including placements, pruning strength, and shaping treatments of ASH, while demonstrating how ASH can be

readily combined with other methods, revealing the unexpected effectiveness and flexibility of such a simple operation (Section 5).

## 2 THE OUT-OF-DISTRIBUTION DETECTION SETUP

OOD detection methods are normally developed with the following recipe:

1. Train a model (e.g. a *classifier*) with some data—namely the **in-distribution (ID)** data. After training, freeze the model parameters.

2. At inference time, feed the model **out-of-distribution (OOD)** data.

3. Turn the model into a *detector* by coming up with a **score** from the model's output, to differentiate whether an input is ID or OOD.

4. Use various **evaluation metrics** to determine how well the detector is doing.

The highlighted keywords are choices to be made in every experimental setting for OOD detection. Following this convention, our design choices for this paper are explained below.

**Datasets and models (Steps 1-2)**    We adopt an experimental setting representative of the previous SOTA: DICE (on CIFAR) and ReAct (on ImageNet). Table 1 summarizes datasets and model architectures used. For CIFAR-10 and CIFAR-100 experiments, we used the 6 OOD datasets adopted in DICE (Sun & Li, 2022): SVHN (Netzer et al., 2011), LSUN-Crop (Yu et al., 2015), LSUN-Resize (Yu et al., 2015), iSUN (Xu et al., 2015), Places365 (Zhou et al., 2017) and Textures (Cimpoi et al., 2014), while the ID dataset is the respective CIFAR. The model used is a pretrained DenseNet-101 (Huang et al., 2017). For ImageNet experiments, we inherit the exact setup from ReAct (Sun et al., 2021), where the ID dataset is ImageNet-1k, and OOD datasets include iNaturalist (Van Horn et al., 2018), SUN (Xiao et al., 2010), Places365 (Zhou et al., 2017), and Textures (Cimpoi et al., 2014). We used ResNet50 (He et al., 2016) and MobileNetV2 (Sandler et al., 2018) network architectures. All networks are pretrained with ID data and never modified post-training; their parameters remain unchanged during the OOD detection phase.

| ID Dataset | OOD Datasets | Model architectures |
|---|---|---|
| CIFAR-10 | SVHN, LSUN C, LSUN R, iSUN, Places365, Textures | DenseNet-101 |
| CIFAR-100 | SVHN, LSUN C, LSUN R, iSUN, Places365, Textures | DenseNet-101 |
| ImageNet | iNaturalist, SUN, Places365, Textures | ResNet50, MobileNetV2 |

Table 1: **Datasets and models in our OOD experiments.** We cover both moderate and large scale OOD benchmark settings, including evaluations on up to 10 OOD datasets and 3 architectures.

**Detection scores (Step 3)**    Commonly used score functions for OOD detection are the maximum/predicted class probability from the Softmax output of the model (Hendrycks & Gimpel, 2017), and the Energy score (Liu et al., 2020). As the Energy score has been shown to outperform Softmax (Liu et al., 2020; Sun et al., 2021), we default to using the former. In our ablation studies we also experimented versions of our method combined with the Softmax score, as well as other methods (see Table 5). For a given input $\mathbf{x}$ and a trained network function $f$, the energy function $E(\mathbf{x}; f)$ maps the logit outputs from the network, $f(\mathbf{x})$, to a scalar: $E(\mathbf{x}; f) = -\log \sum_{i=1}^{C} e^{f_i(\mathbf{x})}$ where $C$ is the number of classes and $f_i(\mathbf{x})$ is the logit output of class $i$. The score used for OOD detection is the negative energy score, $-E(\mathbf{x}; f)$, so that ID samples produce a higher score, consistent with the convention. We compare our method with other scoring methods, e.g. Mahalanobis distance (Lee et al., 2018), as well as other advanced methods that build upon them: ODIN (Liang et al., 2017) (built upon the Softmax score), ReAct (Sun et al., 2021) and DICE (Sun & Li, 2022) (built upon the Energy score).

**Evaluation metrics (Step 4)**    We evaluate our method using threshold-free metrics for OOD detection standardized in Hendrycks & Gimpel (2017): (i) AUROC: the Area Under the Receiver Operating Characteristic curve; (ii) AUPR: Area Under the Precision-Recall curve; and (iii) FPR95:

false positive rate—the probability that a negative (e.g. OOD) example is misclassified as positive (e.g. ID)—when the true positive rate is as high as 95% (Liang et al., 2017). In addition to OOD metrics, we also evaluate each method on their ID performance, which in this case is the classification accuracy on in-distribution data, e.g. Top-1 accuracy on the ImageNet validation set.

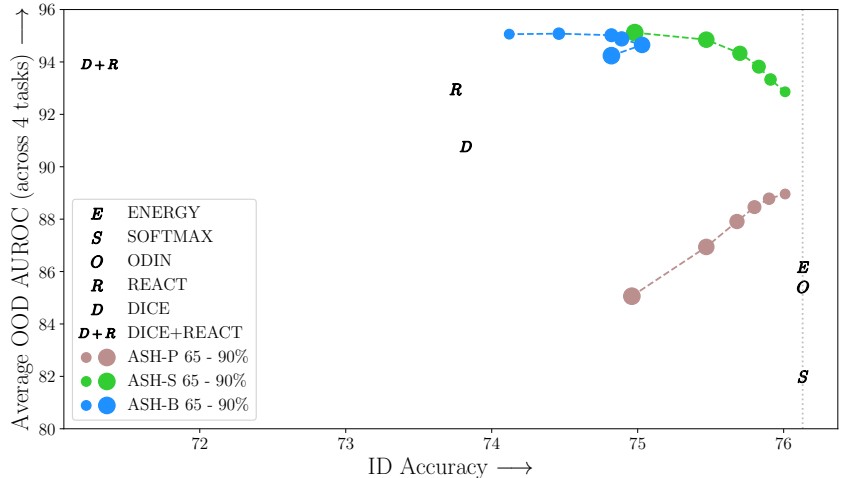

Figure 2: **ID-OOD tradeoff on ImageNet.** Plotted are the average OOD detection rate (AUROC; averaged across 4 OOD datasets - iNaturalist, SUN, Places365, Textures) vs ID classification accuracy (Top-1 accuracy in percentage on ImageNet validation set) of all OOD detection methods and their variants used in this paper. Baseline methods "E", "S" and "O" lie on the upper bound of ID accuracy (indicated by the dotted gray line) since it makes no modification of the network or the features. "R", "D" and "D+R" improve on the OOD metric, but come with an ID accuracy drop. ASH (dots connected with dashed lines; smaller dots indicate lower pruning level) offers the best trade-off and form a Pareto front.

## 3 ACTIVATION SHAPING FOR OOD DETECTION

A trained network converts raw input data (e.g. RGB pixel values) into useful representations (e.g. a stack of spatial activations). We argue that representations produced by modern, over-parameterized deep neural networks are excessive for the task at hand, and therefore could be greatly simplified without much deterioration on the original performance (e.g. classification accuracy), while resulting in a surprising gain on other tasks (e.g. OOD detection). Such a hypothesis is tested via an **a**ctivation **sh**aping method, ASH, which simplifies an input's feature representation with the following recipe:

1. Remove a majority of the activation by obtaining the $p$th-percentile of the entire representation, $t$, and setting all values below $t$ to 0.

2. For the un-pruned activations, apply one of the following treatments:
    - **ASH-P** (Algorithm 1): Do nothing. **P**runing is all we need. This is used as a baseline to highlight the gains of the following two treatments.
    - **ASH-B** (Algorithm 2): Assign all of them to be a positive constant so the entire representation becomes **B**inary.
    - **ASH-S** (Algorithm 3): **S**cale their values up by a ratio calculated from the sum of activation values before and after pruning.

ASH is applied on-the-fly to any input sample's feature representation at an intermediate layer, after which it continues down the forward path throughout the rest of the network, as depicted in Figure 1. Its output—generated from the simplified representation—is then used for both the original task (e.g. classification), or, in the case of OOD detection, obtaining a score, to which a thresholding mechanism is applied to distinguish between ID and OOD samples. ASH is therefore a unified framework for both the original task and OOD detection without incurring any additional computation.

| **Algorithm 1** ASH-P: Activation Shaping with Pruning | **Algorithm 2** ASH-B: Activation Shaping by Binarizing | **Algorithm 3** ASH-S: Activation Shaping with Scaling |
|---|---|---|
| **Input:** Single sample activation $\mathbf{x}$, pruning percentile $p$ | **Input:** Single sample activation $\mathbf{x}$, pruning percentile $p$ | **Input:** Single sample activation $\mathbf{x}$, pruning percentile $p$ |
| **Output:** Modified activation $\mathbf{x}$ | **Output:** Modified activation $\mathbf{x}$ | **Output:** Modified activation $\mathbf{x}$ |
| 1: Calculate the $p$-th percentile of the input $\mathbf{x} \rightarrow$ threshold $t$ | 1: Calculate the $p$-th percentile of the input $\mathbf{x} \rightarrow$ threshold $t$ | 1: Calculate the $p$-th percentile of the input $\mathbf{x} \rightarrow$ threshold $t$ |
|  | 2: Calculate the sum of $\mathbf{x} \rightarrow s$ | 2: Calculate the sum of $\mathbf{x} \rightarrow s1$ |
| 2: Set all values in $\mathbf{x}$ less than $t$ to 0 | 3: Set all values in $\mathbf{x}$ less than $t$ to 0 | 3: Set all values in $\mathbf{x}$ less than $t$ to 0 |
|  | 4: Calculate number of non-zeros in $\mathbf{x} \rightarrow n$ | 4: Calculate the sum of $\mathbf{x} \rightarrow s2$ |
|  | 5: Set all non-zero values in $\mathbf{x}$ to $s/n$ | 5: Multiply all non-zero values in $\mathbf{x}$ with $\exp(s1/s2)$ |
| 3: **return** $\mathbf{x}$ | 6: **return** $\mathbf{x}$ | 6: **return** $\mathbf{x}$ |

**Placement of ASH**   We can apply ASH at various places throughout the network, and the performance would differ. The main results shown in this paper are from applying ASH after the last average pooling layer, for ImageNet experiments, where the feature map size is $2048 \times 1 \times 1$ for ResNet50, and $1280 \times 1 \times 1$ for MobileNetV2. For CIFAR experiments conducted with DenseNet-101, we apply ASH after the penultimate layer, where the feature size is $342 \times 1 \times 1$. Ablation studies for other ASH placements are included in Section 5 and Section A in Appendix.

**The $p$ parameter**   ASH algorithms come with only one parameter, $p$: the pruning percentage. In experiments we vary $p$ from 60 to 90 and have observed relatively steady performances (see Figure 2). When studying its effect on ID accuracy degradation, we cover the entire range from 0 to 100 (see Figure 3). The SOTA performances are given by surprisingly high values of $p$. For ImageNet, the best performing ASH versions are ASH-B with $p = 65$, and ASH-S with $p = 90$. For CIFAR-10 and CIFAR-100, the best performing ASH versions are ASH-S with $p = 95$ and $p = 90$, and comparably, ASH-B with $p = 95$ and $p = 85$. See Section F in Appendix for full details on the parameter choice.

## 4   RESULTS

### 4.1   ASH OFFERS THE BEST ID-OOD TRADEOFF

ASH as a unified pipeline for both ID and OOD tasks demonstrates strong performance at both. Figure 2 shows the ID-OOD tradeoff for various methods for ImageNet. On one end, methods that rely on obtaining a score straight from the unmodified output of the network, e.g. Energy, Softmax and ODIN (Liang et al., 2017), perfectly preserves ID accuracy, but perform relatively poorly for OOD detection. Advanced methods that modify the network weights or representations, e.g. ReAct (Sun et al., 2021) and DICE (Sun & Li, 2022), come at a compromise of ID accuracy when applied as a unified pipeline. In the case of ReAct, the ID accuracy drops from 76.13% to 73.75%.

ASH offers the best of both worlds: optimally preserve ID performance while improving OOD detection. Varying the pruning percentage $p$, we can see in Figure 2 that those ASH-B and ASH-S variants establish a new Pareto frontier. The pruning-only ASH-P gives the same ID accuracy as ASH-S at each pruning level, while falling behind ASH-S on the OOD metric, suggesting that simply by scaling up un-pruned activations, we observe a great performance gain at OOD detection.

### 4.2   OOD DETECTION ON BOTH IMAGENET AND CIFAR BENCHMARKS

ASH is highly effective at OOD detection. For ImageNet, while Figure 2 displays the averaged performance across 4 datasets, Table 2 lays out detailed performances on each of them, with each of the two metrics: FPR95 and AUROC. The table follows the exact format as Sun et al. (2021), reporting results from competitive OOD detection methods in the literature, with additional baselines computed by us (e.g. DICE and DICE+ReAct on MobileNet). As we can see, the proposed ASH-B and ASH-S establish the new SOTA across almost all OOD datasets and evaluation metrics on ResNet, and perform comparably with DICE+ReAct while much more algorithmically simpler.

ASH-P showcases surprising gains from pruning only (simply removing 60% low value activations), outperforming Energy score, Softmax score and ODIN.

On CIFAR benchmarks, we followed the exact experimental setting as Sun & Li (2022): 6 OOD datasets with a pretrained DenseNet-101. Table 3 reports our method's performance (averaged across all 6 datasets) alongside all baseline methods. Detailed per-dataset performance is reported in Table 7 and Table 8 in Appendix. All ASH variants profoundly outperform existing baselines.

| Model | Methods | OOD Datasets | | | | | | | | | |
| | | iNaturalist | | SUN | | Places | | Textures | | Average | |
| | | FPR95 ↓ | AUROC ↑ | FPR95 ↓ | AUROC ↑ | FPR95 ↓ | AUROC ↑ | FPR95 ↓ | AUROC ↑ | FPR95 ↓ | AUROC ↑ |
|---|---|---|---|---|---|---|---|---|---|---|---|
| ResNet | Softmax score | 54.99 | 87.74 | 70.83 | 80.86 | 73.99 | 79.76 | 68.00 | 79.61 | 66.95 | 81.99 |
| | ODIN | 47.66 | 89.66 | 60.15 | 84.59 | 67.89 | 81.78 | 50.23 | 85.62 | 56.48 | 85.41 |
| | Mahalanobis | 97.00 | 52.65 | 98.50 | 42.41 | 98.40 | 41.79 | 55.80 | 85.01 | 87.43 | 55.47 |
| | Energy score | 55.72 | 89.95 | 59.26 | 85.89 | 64.92 | 82.86 | 53.72 | 85.99 | 58.41 | 86.17 |
| | ReAct | 20.38 | 96.22 | 24.20 | 94.20 | 33.85 | 91.58 | 47.30 | 89.80 | 31.43 | 92.95 |
| | DICE | 25.63 | 94.49 | 35.15 | 90.83 | 46.49 | 87.48 | 31.72 | 90.30 | 34.75 | 90.77 |
| | DICE + ReAct | 18.64 | 96.24 | 25.45 | 93.94 | 36.86 | 90.67 | 28.07 | 92.74 | 27.25 | 93.40 |
| | ASH-P (Ours) | 44.57 | 92.51 | 52.88 | 88.35 | 61.79 | 85.58 | 42.06 | 89.70 | 50.32 | 89.04 |
| | ASH-B (Ours) | 14.21 | 97.32 | 22.08 | 95.10 | 33.45 | 92.31 | 21.17 | 95.50 | 22.73 | 95.06 |
| | ASH-S (Ours) | 11.49 | 97.87 | 27.98 | 94.02 | 39.78 | 90.98 | 11.93 | 97.60 | 22.80 | 95.12 |
| MobileNet | Softmax score | 64.29 | 85.32 | 77.02 | 77.10 | 79.23 | 76.27 | 73.51 | 77.30 | 73.51 | 79.00 |
| | ODIN | 55.39 | 87.62 | 54.07 | 85.88 | 57.36 | 84.71 | 49.96 | 85.03 | 54.20 | 85.81 |
| | Mahalanobis | 62.11 | 81.00 | 47.82 | 86.33 | 52.09 | 83.63 | 92.38 | 33.06 | 63.60 | 71.01 |
| | Energy score | 59.50 | 88.91 | 62.65 | 84.50 | 69.37 | 81.19 | 58.05 | 85.03 | 62.39 | 84.91 |
| | ReAct | 42.40 | 91.53 | 47.69 | 88.16 | 51.56 | 86.64 | 38.42 | 91.53 | 45.02 | 89.47 |
| | DICE | 43.09 | 90.83 | 38.69 | 90.46 | 53.11 | 85.81 | 32.80 | 91.30 | 41.92 | 89.60 |
| | DICE + ReAct | 32.30 | 93.57 | 31.22 | 92.86 | 46.78 | 88.02 | 16.28 | 96.25 | 31.64 | 92.68 |
| | ASH-P (Ours) | 54.92 | 90.46 | 58.61 | 86.72 | 66.59 | 83.47 | 48.48 | 88.72 | 57.15 | 87.34 |
| | ASH-B (Ours) | 31.46 | 94.28 | 38.45 | 91.61 | 51.80 | 87.56 | 20.92 | 95.07 | 35.66 | 92.13 |
| | ASH-S (Ours) | 39.10 | 91.94 | 43.62 | 90.02 | 58.84 | 84.73 | 13.12 | 97.10 | 38.67 | 90.95 |

Table 2: **OOD detection results on ImageNet.** We follow the exact same metrics and table format as Sun et al. (2021). Both ResNet and MobileNet are trained with ID data (ImageNet-1k) only. ↑ indicates larger values are better and ↓ indicates smaller values are better. All values are percentages. "DICE" and "DICE+ReAct" for MobileNet are implemented by us (refer to Section F for hyperparameter choices). The rest of the table except for those indicated "Ours" are taken directly from Table 1 in Sun et al. (2021). For ResNet, ASH consistently perform better than benchmarks, across all the OOD datasets. In the case of MobileNet, ASH performs comparably with DICE+ReAct.

| Method | CIFAR-10 | | CIFAR-100 | |
| | FPR95 ↓ | AUROC ↑ | FPR95 ↓ | AUROC ↑ |
|---|---|---|---|---|
| Softmax score | 48.73 | 92.46 | 80.13 | 74.36 |
| ODIN | 24.57 | 93.71 | 58.14 | 84.49 |
| Mahalanobis | 31.42 | 89.15 | 55.37 | 82.73 |
| Energy score | 26.55 | 94.57 | 68.45 | 81.19 |
| ReAct | 26.45 | 94.95 | 62.27 | 84.47 |
| DICE | $20.83^{\pm1.58}$ | $95.24^{\pm0.24}$ | $49.72^{\pm1.69}$ | $87.23^{\pm0.73}$ |
| ASH-P (Ours) | 23.45 | 95.22 | 64.53 | 82.71 |
| ASH-B (Ours) | 20.23 | 96.02 | 48.73 | 88.04 |
| ASH-S (Ours) | 15.05 | 96.61 | 41.40 | 90.02 |

Table 3: **OOD detection results on CIFAR benchmarks.** ↑ indicates larger values are better and ↓ indicates smaller values are better. All values are percentages. Results are averaged accross 6 OOD datasets. Methods except for ASH variants (marked as "Ours") are taken from Sun & Li (2022).

## 4.3 ON PRESERVING IN-DISTRIBUTION ACCURACY

ASH preserves in-distribution performance, and hence can be used as a unified pipeline for ID and OOD tasks. As we can see in Figure 2, towards the low end of pruning percentage (65%) both ASH-S and ASH-P come with only a slight drop of ID accuracy (ImageNet Top-1 validation accuracy;

| Method | ImageNet benchmark | | | |
|--------|------|-------|------|--------|
| | FPR95 ↓ | AUROC ↑ | AUPR ↑ | ID ACC ↑ |
| ASH-RAND@65 | 45.37 | 90.80 | 98.09 | 72.16 |
| ASH-RAND@70 | 46.93 | 90.67 | 98.05 | 72.87 |
| ASH-RAND@75 | 46.93 | 90.67 | 98.05 | 73.19 |
| ASH-RAND@80 | 51.24 | 89.94 | 97.89 | 73.57 |
| ASH-RAND@90 | 59.35 | 87.88 | 97.44 | 73.51 |
| ASH-B@65 | 22.73 | 95.06 | 98.94 | 74.12 |
| ASH-S@90 | 22.80 | 95.12 | 98.90 | 74.98 |

Table 4: **An extreme variant of ASH: randomized activations**. ASH-RAND sets un-pruned activations to random values between 0 and 10. All experiments are on ImageNet with ResNet-50, and the OOD performance is averaged across 4 datasets. ASH-RAND is surprisingly comparable to, although not better than, ASH-B and ASH-S. It consistently beats simple baselines like Energy, Softmax and ODIN.

76.13% to 76.01%). The more we prune, the larger drop of accuracy is seen[1]. At 90%—that is, when 90% of activation values are eliminated—ASH-S and ASH-P maintain an ID accuracy of 74.98%. When experimenting with a wider range and granular pruning levels, as shown in Figure 3, we observe that ASH-S and ASH-P do preserve accuracy all the way, until a rather high value of pruning (e.g. ID accuracy dropped to 64.976% at 99% of pruning).

However, a reversed trend is observed in ASH-B: as seen in Figure 3, between 50-90% of pruning, the ID accuracy is trending up, while the best accuracy is achieved between 80% to 90% of pruned activation. The reason is that the rather extreme binarizing operation in ASH-B (setting all remaining activations to a constant) has a bigger impact when the pruning rate is lower (more values are being modified). To the extreme of 0% pruning, ASH-B simply sets all activation values to their average, which completely destroys the classifier (Figure 3, left end of curve).

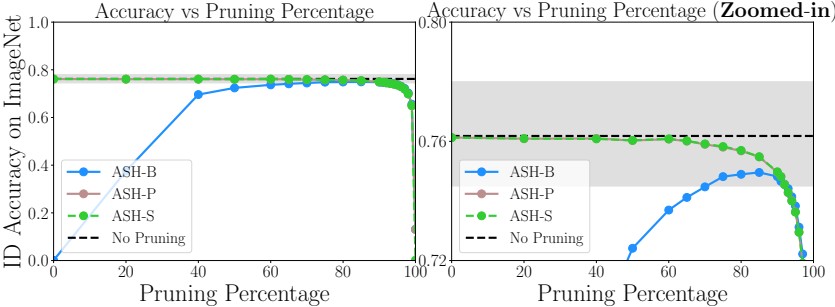

Figure 3: **Accuracy degradation across pruning percentage.** All three versions of ASH are applied to the penultimate layer of a ResNet-50 pretrained on ImageNet. At test time the input samples from the ImageNet validation set are being processed with ASH, and the Top-1 accuracy is reported across a range of pruning strengths. ASH-P and ASH-S have the exact same effect on ID accuracy, as expected. ASH-B fails when pruning percentage is low, as the majority of the feature map would be converted to a constant value. The right plot is a zoomed-in version (on y axis) of the left.

## 4.4 ASH-RAND: RANDOMIZING ACTIVATION VALUES

Given the success of ASH on both ID and OOD tasks, especially ASH-B where the values of a whole feature map are set to either 0 or a positive constant, we are curious to push the extent of activation shaping even further. We experiment with a rather extreme variant: ASH-RAND, which sets the remaining activation values to a random nonnegative value between $[0, 10]$ after pruning. The result is shown in Table 4. It works reasonably well even for such an extreme modification of feature maps.

The strong results of ASH prompt us to ask: why does making radical changes to a feature map improve OOD? Why does pruning away a large portion of features not affect accuracy? Are representations redundant to start with? We discuss useful interpretations of ASH in Appendix Section E.

---

[1]ASH-P and ASH-S produce the exact accuracy on ID tasks, as linearly scaling the last-layer activation and hence logits, barring numerical instability, does not affect a model's softmax output.

# 5 ABLATION STUDIES

**Global vs local threshold** The working version of ASH calculates the pruning threshold $t$ (according to a fixed percentile $p$; see Algorithms 1-3) per-image and on the fly, that is, each input image would have a pruning step applied with a different threshold. This design choice requires no global information about the network, training or test data, but incurs a slight computational overhead at inference. An alternative is to calculate a "global" threshold from all training data, assuming we have access to them post training. There a 90% pruning level means gathering statistics of a certain feature map from all training data and obtaining a value that reflect the 90% percentile.

We implement both ASH-S and ASH-B with both global and local thresholds; the difference between the two design choices are showcased in Figure 4 for ASH-S. ASH-B results are included in Section D in Appendix. As we can see, with aligned pruning percentage, using a local threshold always works better than setting a global one, while the best overall performance is also achieved by local thresholds.

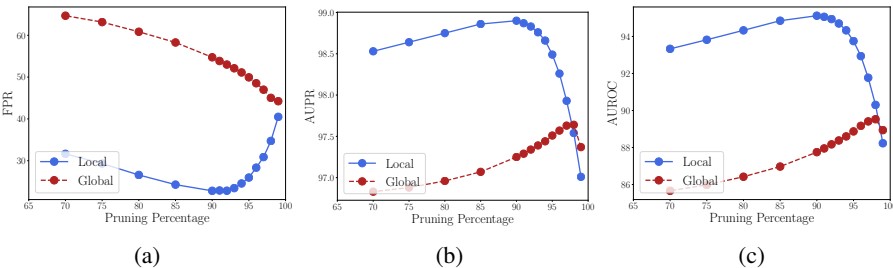

Figure 4: **Global vs local threshold.** We implement a variant of ASH with global thresholds obtained from training data, and compare the results with the defaulted local threshold version. Shown are ASH-S with a ResNet-50 trained on ImageNet, with local vs global curves on FPR95, AUROC, and AUPR, respectively. Local thresholds, while incurring a slight overhead at inference, perform much better and require no access to training data.

**Where to ASH** We notice that a working placement of ASH is towards later layers of a trained network, e.g. the penultimate layer. We experimented how other placements affect its performance. In Figure 5 we show the effect of performing ASH-S on different layers of a network. We can see that the accuracy deterioration over pruning rate becomes more severe as we move to earlier parts of the network. For full OOD detection results from these placements, refer to Section A in Appendix.

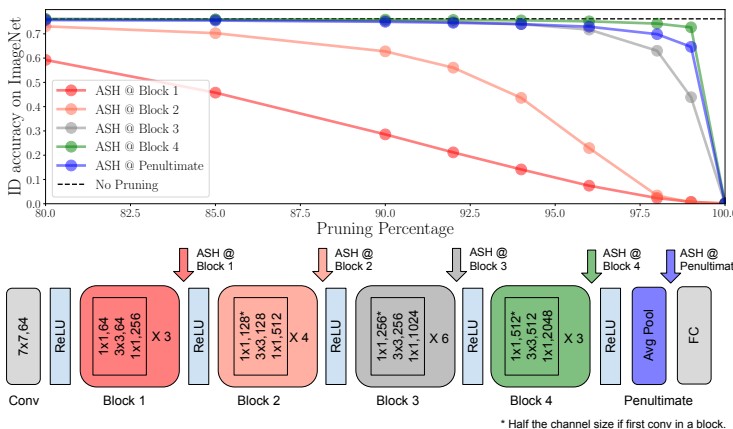

Figure 5: **Effect of ASH placements on ID Accuracy.** (Top) ID accuracy degradation curves across pruning percentage, each color indicating a different placement of ASH. (Bottom) Diagram of a ResNet-50 architecture and indications of where ASH is applied.

**Plug-and-play ASH to other methods** ASH as a two-step operation is very much compatible with other existing methods. In Table 5 we demonstrate how it can be easily combined with the Softmax

score, the Energy score, ODIN and ReAct to provide immediate improvement over them. Experiments are performed with CIFAR-10, CIFAR-100 and ImageNet. All methods are reimplemented by us. See Section F in Appendix for implementation details.

| Method | CIFAR-10 | | | CIFAR-100 | | | ImageNet | | |
|---|---|---|---|---|---|---|---|---|---|
| | FPR95 ↓ | AUROC ↑ | AUPR ↑ | FPR95 ↓ | AUROC ↑ | AUPR ↑ | FPR95 ↓ | AUROC ↑ | AUPR ↑ |
| Softmax score | 48.69 | 92.52 | 80.75 | 80.06 | 74.45 | 76.99 | 64.76 | 82.82 | 95.94 |
| Softmax score + ASH tr. | 48.86 | 92.61 | 77.03 | 76.04 | 75.00 | 78.61 | 37.86 | 90.90 | 97.97 |
| ODIN | 25.71 | 94.72 | 95.60 | 64.87 | 82.43 | 84.85 | 50.80 | 87.57 | 97.19 |
| ODIN + ASH tr. | 15.38 | 96.41 | 96.62 | 38.54 | 90.49 | 91.51 | 28.59 | 93.34 | 98.40 |
| Energy score | 26.59 | 94.63 | 95.61 | 68.29 | 81.23 | 83.64 | 57.47 | 87.05 | 97.15 |
| Energy score + ASH tr. | 15.05 | 96.61 | 96.88 | 41.40 | 90.02 | 91.23 | 22.80 | 95.12 | 98.90 |
| ReAct | 29.00 | 94.92 | 96.14 | 69.94 | 82.07 | 85.43 | 31.43 | 92.95 | 98.50 |
| ReAct + ASH tr. | 16.35 | 96.91 | 97.41 | 41.64 | 88.93 | 90.14 | 24.88 | 94.27 | 98.66 |

Table 5: **The ASH treatment (ASH tr.) is compatible with and improves on existing methods.** CIFAR10 and CIFAR100 results are averaged across 6 different OOD tasks and ImageNet results are averaged across 4 different OOD tasks. All methods and experiments are implemented by us. Note that "Energy score + ASH tr." is the same as what's referred to as "ASH" elsewhere.

## 6 RELATED WORK

**Post-hoc model enhancement**   Our method makes post-hoc modifications to trained representations of data. Similar practices have been incorporated in other domains. For example, Monte Carlo dropout (Gal & Ghahramani, 2016) estimates predictive uncertainty by adding dropout layers at both training and test time, generating multiple predictions of the same input instance. In adversarial defense, randomized smoothing (Cohen et al., 2019) applies random Gaussian perturbations of an input to obtain robust outputs. The downside of these methods is that they all require multiple inference runs per input image. The field of model editing (Santurkar et al., 2021; Meng et al., 2022) and fairness (Alabdulmohsin & Lucic, 2021; Celis et al., 2019) also modify trained models for gains in robustness and fairness guarantees. However, they all involve domain expertise as well as additional data, neither of which required in our work. Temperature scaling (Guo et al., 2017) rescales the neural network's logits by a scalar learned from a separate validation dataset. ODIN (Liang et al., 2017) then combines temperature scaling with input perturbation. We have compared closely with ODIN in our experiments and have shown superior results.

**Sparse representations**   A parallel can be drawn between ASH and activation pruning, and the more general concept of sparse representations. Stochastic activation pruning (SAP) (Dhillon et al., 2018) has been proposed as a useful technique against adversarial attacks. SAP prunes a random subset of low-magnitude activations during each forward pass and scales up the others. Adversarial defense is out of scope for this paper, but serves as a fruitful future work direction. Ahmad & Scheinkman (2019) used combinatorics of random vectors to show how sparse activations might enable units to use lower thresholds, which can have a number of benefits like increased noise tolerance. They use top-K in their network as an activation function, in place of ReLU. The key difference from ASH is the usage of top-K to create sparse activations even during training, and the removal of ReLU.

## 7 CONCLUSION

In this paper, we present ASH, an extremely simple, post hoc, on-the-fly, and plug-and-play activation shaping method applied to inputs at inference. ASH works by pruning a large portion of an input sample's activation and lightly adjusting the remaining. When combined with the energy score, it's shown to outperform all contemporary methods for OOD detection, on both moderate and large-scale image classification benchmarks. It is also compatible with and provides benefits to existing methods. The extensive experimental setup on 3 ID datasets, 10 OOD datasets, and performances evaluated on 4 metrics, demonstrates the effectiveness of ASH across the board: reaching SOTA on OOD detection while providing the best trade-off between OOD detection and ID classification accuracy.

ACKNOWLEDGEMENTS

This work was supported by the ML Collective compute grant funded by Google Cloud, and was selected by the ICLR 2022 DEI initiative (Liu & Maughan, 2021; 2022) as a sponsored project. The authors would like to thank Marcus Lewis for proofreading and supplying connections to the sparsity literature, Dumitru Erhan for giving feedback on early drafts, and Milan Misic for fruitful discussions and brainstorming in the early phase of the project. We thank Fargo the dog for his canine countenance. We are grateful for the ML Collective community and the ICLR CoSubmitting Summer community for the ongoing support and feedback of this research.

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

# APPENDIX: EXTREMELY SIMPLE ACTIVATION SHAPING FOR OUT-OF-DISTRIBUTION DETECTION

## A  FULL RESULTS ON ASH PLACEMENTS

The main results shown in the paper are generated by placing ASH at later layers of the network: the penultimate layer for ResNet-50 and MobileNet. Here we show results if we place ASH at different locations of the network: 1st, 2nd, 3rd and 4th Block of ResNet-50, at the end of the last convolution layer at each block before the activation function (ReLU). Figure 5 (bottom) illustrates such placements, along with accuracy degradation curves (top). Here we show the OOD detection results from these ASH placements in Table 6.

As we can see, indeed the penultimate layer placement gives the best result in terms of ID accuracy preservation and OOD detection. As we move ASH towards the beginning of the network, both accuracy and OOD detection rates degrade.

It is worth noting that results in Table 6 adopt a different version of ASH for each layer block. For the penultimate layer and 4th Layer, we use ASH-S@90 (the same setting as that in the main paper), while for 1-3 Layers we use ASH-P@90. The reason is that ASH-S drastically degrades the performance. By simply adding a scaling factor (changing from ASH-P to ASH-S) on Layer 3, the ID accuracy drops from 75% to 5%.

|  | ID: ImageNet; OOD: iNaturalist, Places, Textures, Sun | | | |
| --- | --- | --- | --- | --- |
| ASH placement | FPR95 ↓ | AUROC ↑ | AUPR ↑ | ID ACC ↑ |
| ASH-P@90 before last ReLU of 1st Block | 93.55 | 59.46 | 89.89 | 28.57 |
| ASH-P@90 before last ReLU of 2nd Block | 70.45 | 81.45 | 95.93 | 62.78 |
| ASH-P@90 before last ReLU of 3rd Block | 63.38 | 85.83 | 96.91 | 75.36 |
| ASH-S@90 before last ReLU of 3rd Block | 98.51 | 41.39 | 83.59 | 5.21 |
| ASH-S@70 before last ReLU of 3rd Block | 97.72 | 49.50 | 87.27 | 19.32 |
| ASH-S@90 before last ReLU of 4th Block | 34.69 | 92.11 | 98.38 | 75.83 |
| ASH-B@90 before last ReLU of 4th Block | 33.74 | 92.37 | 98.48 | 75.70 |
| ASH-S@90 after penultimate Layer* | 22.80 | 95.12 | 98.90 | 74.98 |
| No ASH (Energy score alone) | 58.41 | 86.17 | 96.88 | 76.13 |

Table 6: **ASH applied to different places throughout a ResNet-50, trained on ImageNet**. OOD results are averaged across 4 different datasets/tasks: iNaturalist, Places, Textures, Sun. ↑ indicates larger values are better and ↓ indicates smaller values are better. All values are percentages. Experimental setup is described in Section 2

## B  DETAILED CIFAR RESULTS

Table 7 and Table 8 supplement Table 2 in the main text, as they display the full results on each of the 6 OOD datasets for models trained on CIFAR-10 and CIFAR-100 respectively.

## C  ID-OOD TRADEOFF FOR ADDITIONAL ARCHITECTURES

While Figure 2 depicts the ID-OOD tradeoff for ImageNet dataset on ResNet-50 architecture, we supplement the figure with an additional architecture: MobileNetV2, whose OOD results are included in Table 2. As we can see in Figure 6, both ASH-S (green) and ASH-B (blue) lines offer superior ID accuracy preservation, while offering comparable OOD performances with state of the art (DICE+ReAct). Note that while on average, for this particular architecture, ASH is not outperforming DICE+ReAct on OOD metrics, it still offers a number of advantages: algorithmically simpler, much lighter turning effort, zero precomputing cost, and preserving ID accuracy.

Table 7: Detailed results on six common OOD benchmark datasets: Textures (Cimpoi et al., 2014), SVHN (Netzer et al., 2011), Places365 (Zhou et al., 2017), LSUN-Crop (Yu et al., 2015), LSUN-Resize (Yu et al., 2015), and iSUN (Xu et al., 2015). For each ID dataset, we use the same DenseNet pretrained on **CIFAR-10**. ↑ indicates larger values are better and ↓ indicates smaller values are better.

| Method | SVHN | | LSUN-c | | LSUN-r | | iSUN | | Textures | | Places365 | | Average | |
|---|---|---|---|---|---|---|---|---|---|---|---|---|---|---|
| | FPR95 ↓ | AUROC ↑ | FPR95 ↓ | AUROC ↑ | FPR95 ↓ | AUROC ↑ | FPR95 ↓ | AUROC ↑ | FPR95 ↓ | AUROC ↑ | FPR95 ↓ | AUROC ↑ | FPR95 ↓ | AUROC ↑ |
| Softmax score | 47.24 | 93.48 | 33.57 | 95.54 | 42.10 | 94.51 | 42.31 | 94.52 | 64.15 | 88.15 | 63.02 | 88.57 | 48.73 | 92.46 |
| ODIN | 25.29 | 94.57 | 4.70 | 98.86 | 3.09 | 99.02 | 3.98 | 98.90 | 57.50 | 82.38 | 52.85 | 88.55 | 24.57 | 93.71 |
| GODIN | 6.68 | 98.32 | 17.58 | 95.09 | 36.56 | 92.09 | 36.44 | 91.75 | 35.18 | 89.24 | 73.06 | 77.18 | 34.25 | 90.61 |
| Mahalanobis | 6.42 | 98.31 | 56.55 | 86.96 | 9.14 | 97.09 | 9.78 | 97.25 | 21.51 | 92.15 | 85.14 | 63.15 | 31.42 | 89.15 |
| Energy score | 40.61 | 93.99 | 3.81 | 99.15 | 9.28 | 98.12 | 10.07 | 98.07 | 56.12 | 86.43 | 39.40 | 91.64 | 26.55 | 94.57 |
| ReAct | 41.64 | 93.87 | 5.96 | 98.84 | 11.46 | 97.87 | 12.72 | 97.72 | 43.58 | 92.47 | 43.31 | 91.03 | 26.45 | 94.67 |
| DICE | 25.99±5.10 | 95.90±1.08 | 0.26±0.11 | 99.92±0.02 | 3.91±0.56 | 99.20±0.15 | 4.36±0.71 | 99.14±0.15 | 41.90±4.41 | 88.18±1.80 | 48.59±1.53 | 89.13±0.31 | 20.83±1.58 | 95.24±0.24 |
| ASH-P (Ours) | 30.14 | 95.29 | 2.82 | 99.34 | 7.97 | 98.33 | 8.46 | 98.29 | 50.85 | 88.29 | 40.46 | 91.76 | 23.45 | 95.22 |
| ASH-B (Ours) | 17.92 | 96.86 | 2.52 | 99.48 | 8.13 | 98.54 | 8.59 | 98.45 | 35.73 | 92.88 | 48.47 | 89.93 | 20.23 | 96.02 |
| ASH-S (Ours) | 6.51 | 98.65 | 0.90 | 99.73 | 4.96 | 98.92 | 5.17 | 98.90 | 24.34 | 95.09 | 48.45 | 88.34 | 15.05 | 96.61 |

Table 8: Detailed results on six common OOD benchmark datasets: Textures (Cimpoi et al., 2014), SVHN (Netzer et al., 2011), Places365 (Zhou et al., 2017), LSUN-Crop (Yu et al., 2015), LSUN-Resize (Yu et al., 2015), and iSUN (Xu et al., 2015). For each ID dataset, we use the same DenseNet pretrained on **CIFAR-100**. ↑ indicates larger values are better and ↓ indicates smaller values are better.

| Method | SVHN | | LSUN-c | | LSUN-r | | iSUN | | Textures | | Places365 | | Average | |
|---|---|---|---|---|---|---|---|---|---|---|---|---|---|---|
| | FPR95 ↓ | AUROC ↑ | FPR95 ↓ | AUROC ↑ | FPR95 ↓ | AUROC ↑ | FPR95 ↓ | AUROC ↑ | FPR95 ↓ | AUROC ↑ | FPR95 ↓ | AUROC ↑ | FPR95 ↓ | AUROC ↑ |
| Softmax score | 81.70 | 75.40 | 60.49 | 85.60 | 85.24 | 69.18 | 85.99 | 70.17 | 84.79 | 71.48 | 82.55 | 74.31 | 80.13 | 74.36 |
| ODIN | 41.35 | 92.65 | 10.54 | 97.93 | 65.22 | 84.22 | 67.05 | 83.84 | 82.34 | 71.48 | 82.32 | 76.84 | 58.14 | 84.49 |
| GODIN | 36.74 | 93.51 | 43.15 | 89.55 | 40.31 | 92.61 | 37.41 | 93.05 | 64.26 | 76.72 | 95.33 | 65.97 | 52.87 | 85.24 |
| Mahalanobis | 22.44 | 95.67 | 68.90 | 86.30 | 23.07 | 94.20 | 31.38 | 93.21 | 62.39 | 79.39 | 92.66 | 61.39 | 55.37 | 82.73 |
| Energy score | 87.46 | 81.85 | 14.72 | 97.43 | 70.65 | 80.14 | 74.54 | 78.95 | 84.15 | 71.03 | 79.20 | 77.72 | 68.45 | 81.19 |
| ReAct | 83.81 | 81.41 | 25.55 | 94.92 | 60.08 | 87.88 | 65.27 | 86.55 | 77.78 | 78.95 | 82.65 | 74.04 | 62.27 | 84.47 |
| DICE | 54.65$^{\pm 4.94}$ | 88.84$^{\pm 0.39}$ | 0.93$^{\pm 0.07}$ | 99.74$^{\pm 0.01}$ | 49.40$^{\pm 1.99}$ | 91.04$^{\pm 1.49}$ | 48.72$^{\pm 1.55}$ | 90.08$^{\pm 1.36}$ | 65.04$^{\pm 0.66}$ | 76.42$^{\pm 0.35}$ | 79.58$^{\pm 2.34}$ | 77.26$^{\pm 1.08}$ | 49.72$^{\pm 1.69}$ | 87.23$^{\pm 0.73}$ |
| ASH-P (Ours) | 81.86 | 83.86 | 11.60 | 97.89 | 67.56 | 81.67 | 70.90 | 80.81 | 78.24 | 74.09 | 77.03 | 77.94 | 64.53 | 82.71 |
| ASH-B (Ours) | 53.52 | 90.27 | 4.46 | 99.17 | 48.38 | 91.03 | 47.82 | 91.09 | 53.71 | 84.25 | 84.52 | 72.46 | 48.73 | 88.04 |
| ASH-S (Ours) | 25.02 | 95.76 | 5.52 | 98.94 | 51.33 | 90.12 | 46.67 | 91.30 | 34.02 | 92.35 | 85.86 | 71.62 | 41.40 | 90.02 |

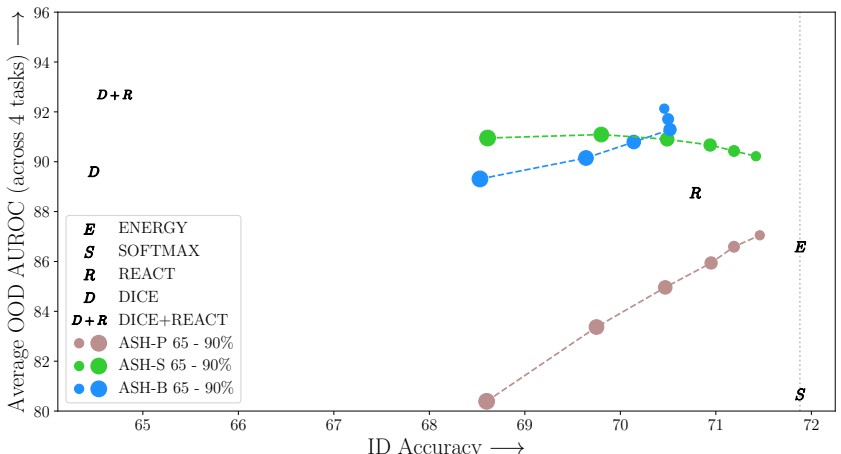

Figure 6: **ID-OOD tradeoff on MobileNet architecture.** Plotted are the average OOD detection rate (AUROC; averaged across 4 OOD datasets - iNaturalist, SUN, Places365, Textures) vs ID classification accuracy (Top-1 accuracy in percentage on ImageNet validation set) of all OOD detection methods and their variants used in this paper on MobileNet architecture. Baseline methods "E" and "S" lie on the upper bound of ID accuracy (indicated by the dotted gray line) since it makes no modification of the network or the features. "R", "D" and "D+R" improve on the OOD metric, but come with an ID accuracy drop. ASH (dots connected with dashed lines; smaller dots indicate lower pruning level) offers the best trade-off and form a Pareto front.

## D    MORE EXPERIMENTS ON GLOBAL VS LOCAL THRESHOLD

Data used to generate Figure 4 can be seen in Table 9. Detailed results comparing local and global thresholds with a range of pruning percentage values and on three metrics. Alternatively, the comparison of global and local thresholds with ASH-B can be found in Table 10.

| Method | Local threshold | | | Global threshold | | |
|---|---|---|---|---|---|---|
| | FPR95 | AUROC | AUPR | FPR95 | AUROC | AUPR |
| | ↓ | ↑ | ↑ | ↓ | ↑ | ↑ |
| ASH-S@99 | 40.49 | 88.23 | 97.01 | 44.24 | 88.94 | 97.37 |
| ASH-S@98 | 34.72 | 90.30 | 97.54 | 45.03 | 89.53 | 97.64 |
| ASH-S@97 | 30.88 | 91.77 | 97.93 | 46.99 | 89.41 | 97.63 |
| ASH-S@96 | 28.34 | 92.94 | 98.26 | 48.53 | 89.17 | 97.57 |
| ASH-S@95 | 25.97 | 93.75 | 98.49 | 49.94 | 88.87 | 97.51 |
| ASH-S@94 | 24.56 | 94.33 | 98.66 | 51.11 | 88.60 | 97.44 |
| ASH-S@93 | 23.45 | 94.70 | 98.76 | 52.11 | 88.38 | 97.39 |
| ASH-S@92 | 22.82 | 94.94 | 98.83 | 53.00 | 88.17 | 97.34 |
| ASH-S@91 | 22.88 | 95.06 | 98.87 | 53.85 | 87.95 | 97.29 |
| ASH-S@90 | 22.80 | 95.12 | 98.90 | 54.74 | 87.75 | 97.25 |
| ASH-S@85 | 24.28 | 94.85 | 98.86 | 58.28 | 86.97 | 97.07 |
| ASH-S@80 | 26.59 | 94.33 | 98.75 | 60.84 | 86.42 | 96.96 |
| ASH-S@75 | 29.32 | 93.82 | 98.64 | 63.16 | 85.98 | 96.88 |
| ASH-S@70 | 31.64 | 93.33 | 98.53 | 64.68 | 85.66 | 96.83 |

Table 9: **Global vs local thresholds: ASH-S**. Shown are ASH-S results with a ResNet-50 trained on ImageNet. Local threshold perform consistently better then global threshold.

| Method | Local threshold | | | Global threshold | | |
|--------|-------|-------|-------|-------|-------|-------|
|        | FPR95 $\downarrow$ | AUROC $\uparrow$ | AUPR $\uparrow$ | FPR95 $\downarrow$ | AUROC $\uparrow$ | AUPR $\uparrow$ |
| ASH-B@99 | 45.43 | 89.09 | 97.55 | 41.70 | 89.75 | 97.55 |
| ASH-B@98 | 39.59 | 91.22 | 98.08 | 41.64 | 90.57 | 97.87 |
| ASH-B@97 | 36.54 | 92.17 | 98.30 | 42.73 | 90.57 | 97.88 |
| ASH-B@96 | 34.26 | 92.79 | 98.44 | 43.55 | 90.43 | 97.85 |
| ASH-B@95 | 32.64 | 93.20 | 98.52 | 44.38 | 90.20 | 97.79 |
| ASH-B@94 | 31.09 | 93.51 | 98.59 | 45.39 | 89.99 | 97.75 |
| ASH-B@93 | 30.03 | 93.76 | 98.64 | 45.77 | 89.86 | 97.72 |
| ASH-B@92 | 29.01 | 93.95 | 98.68 | 46.29 | 89.71 | 97.68 |
| ASH-B@91 | 28.43 | 94.10 | 98.71 | 46.97 | 89.55 | 97.64 |
| ASH-B@90 | 27.58 | 94.24 | 98.74 | 48.25 | 89.39 | 97.61 |
| ASH-B@85 | 25.26 | 94.65 | 98.83 | 51.04 | 88.77 | 97.48 |
| ASH-B@80 | 24.04 | 94.88 | 98.88 | 53.63 | 88.24 | 97.37 |
| ASH-B@75 | 22.95 | 95.02 | 98.91 | 56.66 | 87.63 | 97.25 |
| ASH-B@70 | 22.39 | 95.08 | 98.93 | 60.19 | 86.93 | 97.12 |

Table 10: **Global vs local thresholds: ASH-B**. Shown are ASH-B results with a ResNet-50 trained on ImageNet. Local threshold perform consistently better then global threshold.

## E  INTUITION AND EXPLANATION OF ASH

**ASH as post hoc regularization**    ASH can be thought of as a simple "feature cleaning" step, or a post hoc regularization of features. As we all now acknowledge that NNs are overparameterized, consequently, we can think of the representation learned by such networks as likely "overrepresented." While the power of overparameterization shines mostly through making the training easier—adding more dimensions to the objective landscape while the intrinsic dimension of the task at hand remains constant (Li et al., 2018), we argue that from the lens of representation learning, overparameterized networks "overdo" feature representation, i.e. the representation produced of an input contain too much redundancy. We conjecture that a simple feature cleaning step post training can help ground the resulting learned representation better. Future work to validate, or invalidate this conjecture would include testing if simplified or regularized representation works well in other problem domains, from generalization, to transfer learning and continual learning.

**Connection to a modified ReLU**    Another lens with which to interpret ASH is to see it as a modified ReLu function, adapted on-the-fly per input sample. Since we operate on feature activations pre-ReLU, as shown in Figure 7, the most basic version, ASH-P, combined with the subsequent ReLU fuction, becomes simply an adjusted ReLU. Since the cut-off is determined per-input and on-the-fly, it is essentially an data-dependent activation function. The success of ASH highlights the need to look into more flexible, adaptive, data-dependent activation functions at inference.

**Magnitude vs value in pruning**    Lots of existing pruning methods rely on the *magnitude* of a number (weights or activations). However in this work we use the direct values. That is, a large negative will be pruned if it is within the $p$-percentile of value distribution. The reason is that we operate on activations either *before ReLU*, in which case all negative values will be removed subsequently, or on the penultimate layer where all values are already non-negative.

**How ASH changes the score distribution**    One perspective to interpret the effectiveness of ASH is to examine how it morphs the score distribution to maximize separation between ID and OOD. Since we default to using Energy as the score function, we demonstrate how the energy score distributions change with different strength of ASH treatment, in Figure 8. From no ASH treatment (pruning: 0%) from the left plot to full treatment (90%) on the right, we can see how both ID and OOD distributions are morphed to maximize separation. [2]

---

[2]For animated plots, see: `https://andrijazz.github.io/ash/#dist`

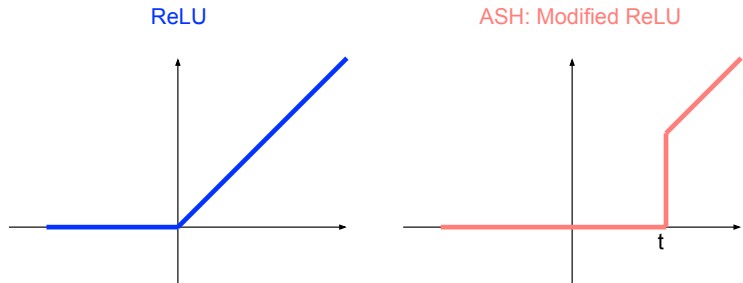

Figure 7: **ASH as a modified ReLU.** Comparison of a regular ReLU activation funciton (left) with a modified ReLU (right), which is equivalent to the ASH-P operation. The pruning threshold $t$ is input data dependent, which makes ASH-P equivalent to a modified ReLU with adaptive thresholds.

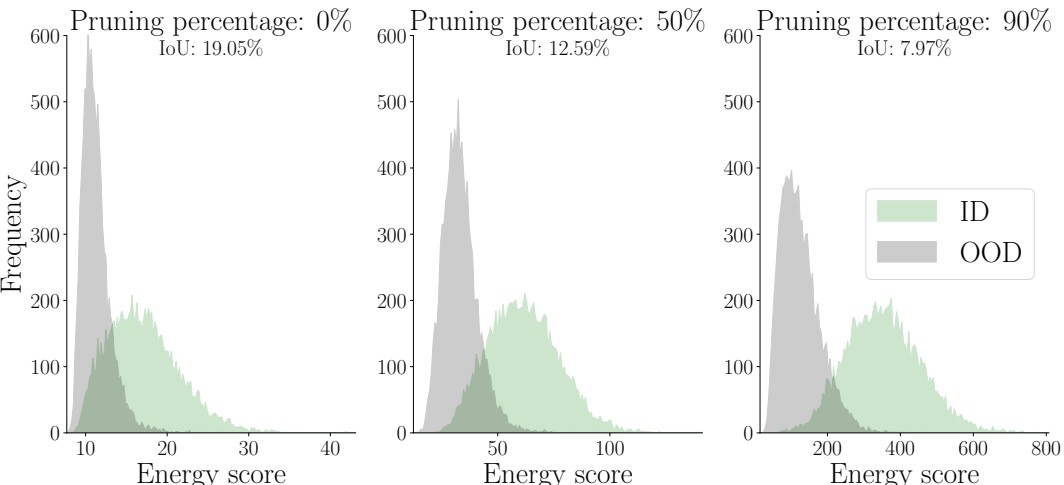

Figure 8: **Energy score distributions.** ASH with increasing pruning strength from left to right morphs the distributions of ID and OOD data, improving the separation, indicated by the Intersection Over Union (IoU) measure. ID data is ImageNet, and OOD data iNaturalist. Shown are energy scores without ASH (left), with ASH-S at 50% pruning strength (middle) and ASH-S at 90% (right).

## F  NOTES ON IMPLEMENTATION

All experiments are done with NVIDIA GTX1080Ti GPUs. Code to reproduce results is submitted alongside this appendix.

Notes on ImageNet results (Table 2):

- For both ResNet and MobileNet results, "ASH-P (Ours)" is implemented with $p = 60$, "ASH-B (Ours)" is implemented with $p = 65$, and "ASH-S (Ours)" is implemented with $p = 90$.

- For the reimplementation of DICE with MobileNet, we used a DICE pruning threshold of 70%.

- For reimplementing DICE + ReAct on MobileNet, since DICE and ReAct each come with their own hyperparameter, we tried a grid search, where DICE pruning thresholds include $\{10\%, 15\%, 70\%\}$ and ReAct clipping thresholds $\{1.0, 1.5, 1.33\}$. Rationals for choosing those values for grid search are: 10% and 15% are taken from the hyperparameter setup in

the DICE codebase[3], and 70% is the recommended threshold of DICE when used alone. In the case of ReAct clipping thresholds, 1.0 and 1.5 are taken from the same codebase, while 1.33 is the 90% percentile calculated from training data, following the ReAct procedure. We report the best result out of all the hyperparameter combinations, given by 10% DICE pruning and 1.0 ReAct clipping.

Notes on CIFAR results (Table 3, Table 7, Table 8):

- For CIFAR-10, 'ASH-P (Ours)" is implemented with $p = 90$, "ASH-B (Ours)" is implemented with $p = 95$, and "ASH-S (Ours)" is implemented with $p = 95$.
- For CIFAR-100, 'ASH-P (Ours)" is implemented with $p = 80$, "ASH-B (Ours)" is implemented with $p = 85$, and "ASH-S (Ours)" is implemented with $p = 90$.
- Results for other methods are copied from DICE.

Notes on compatibility results (Table 5):

- "Softmax score + ASH" is implemented with ASH-P @ $p = 95$ for CIFAR-10, ASH-B @ $p = 70$ for CIFAR-100, and ASH-B @ $p = 65$ for ImageNet.
- "Energy score + ASH" is implemented with ASH-S @ $p = 95$ for CIFAR-10, ASH-S @ $p = 90$ for CIFAR-100, and ASH-S @ $p = 90$ for ImageNet.
- "ODIN" and "ODIN + ASH" use a pretrained DenseNet-101 for CIFAR, where the magnitude parameter is 0.0028, and a pretrained ResNet-50, where the magnitude parameter is 0.005.
- "ODIN + ASH" is implemented with ASH-S @ $p = 95$ for CIFAR-10, ASH-S @ $p = 90$ for CIFAR-100, and ASH-S @ $p = 90$ for ImageNet.
- "ReAct" results for CIFAR-10 and CIFAR-100 are implemented by us. We are unable to replicate the exact results shown in the DICE paper supplementary Table 9.
- "ReAct + ASH" is implemented with ASH-S @ $p = 90$ and ReAct clippign threshold of 1.0 for all of CIFAR-10, CIFAR-100, and ImageNet.

## G    ADDITIONAL ARCHITECTURES

In Table 2 we used 2 architectures (ResNet50 and MobileNetV2) for the ImageNet experiment. We included 2 more here in Table 11: VGG16 (Simonyan & Zisserman, 2014) and DenseNet-121 (Huang et al., 2017).

| Model | Methods | OOD Datasets | | | | | | | | | |
| | | iNaturalist | | SUN | | Places | | Textures | | Average | |
| | | FPR95 ↓ | AUROC ↑ | FPR95 ↓ | AUROC ↑ | FPR95 ↓ | AUROC ↑ | FPR95 ↓ | AUROC ↑ | FPR95 ↓ | AUROC ↑ |
|---|---|---|---|---|---|---|---|---|---|---|---|
| | Energy score | 39.69 | 92.66 | 51.98 | 87.40 | 57.84 | 85.17 | 52.11 | 85.42 | 50.40 | 87.66 |
| | **ASH-S@90 (Ours)** | **15.53** | **97.03** | 37.14 | 91.53 | 46.50 | 88.79 | **22.04** | 95.01 | **30.30** | 93.09 |
| DenseNet-121 | **ASH-B@65 (Ours)** | 34.05 | 92.28 | 42.98 | 89.11 | 55.09 | 84.90 | 56.21 | 83.10 | 47.08 | 87.35 |
| | **ASH-B@90 (Ours)** | 18.22 | 96.36 | **35.17** | **92.48** | **45.38** | **89.15** | 22.75 | **95.38** | 30.38 | **93.34** |
| | Energy score | 51.35 | 90.30 | 57.54 | 87.55 | 64.20 | 84.83 | 44.24 | 89.98 | 54.33 | 88.17 |
| | **ASH-B@65 (Ours)** | **25.98** | **94.20** | **30.04** | 93.19 | **42.52** | **89.01** | **30.25** | **92.35** | **32.20** | **92.19** |
| VGG-16 | **ASH-S@90 (Ours)** | 47.14 | 91.34 | 55.20 | 88.41 | 61.58 | 85.92 | 44.43 | 89.96 | 52.09 | 88.91 |
| | **ASH-S@95 (Ours)** | 38.36 | 92.98 | 47.41 | **94.02** | 54.63 | 87.81 | 39.08 | 90.95 | 44.87 | 90.47 |
| | **ASH-S@99 (Ours)** | 29.30 | 93.18 | 43.59 | 90.17 | 49.46 | 87.79 | 43.60 | 86.36 | 41.49 | 89.38 |

Table 11: **OOD detection results with other architectures on ImageNet.** We follow the exact same metrics and table format as Sun et al. (2021). Both DenseNet-121 and VGG-16 are pretrained on ImageNet-1k. ↑ indicates larger values are better and ↓ indicates smaller values are better. All values are percentages. For DenseNet-121, ASH-S@90 and ASH-B@65 have the best results. In the case of VGG-16, ASH-B@65 consistently perform better than benchmarks, across all the OOD datasets.

---

[3]https://github.com/deeplearning-wisc/dice/blob/4d393c2871a80d8789cc97c31adcee879fe74b29/demo-imagenet.sh

## H   ADDITIONAL SCALING FUNCTIONS

ASH-S can be used with different scaling functions. Table 12 shows comparison of ASH-S performance when used with Linear ($f(x) = x \cdot \frac{s_1}{s_2}$) and Exponential scaling ($f(x) = x \cdot \exp^{\frac{s_1}{s_2}}$) functions. We observed best performance when used with Exponential function.

| | OOD Datasets | | | | | | | | | |
|---|---|---|---|---|---|---|---|---|---|---|
| Scaling functions | iNaturalist | | SUN | | Places | | Textures | | Average | |
| | FPR95 ↓ | AUROC ↑ | FPR95 ↓ | AUROC ↑ | FPR95 ↓ | AUROC ↑ | FPR95 ↓ | AUROC ↑ | FPR95 ↓ | AUROC ↑ |
| Linear (ASH-S@90) | 23.68 | 95.97 | 44.03 | 90.80 | 54.92 | 87.90 | 24.24 | 94.55 | 36.72 | 92.31 |
| Exponential (ASH-S@90) | 11.49 | 97.87 | 27.98 | 94.02 | 39.78 | 90.98 | 11.93 | 97.60 | 22.80 | 95.12 |

Table 12: **Comparison of different scaling functions used with ASH-S.**

## I   CIFAR 10 VS CIFAR 100

We used CIFAR 10 and CIFAR 100 as ID datasets in our experiments, while using six other, rather different datasets (SVHN, LSUN C, LSUN R, iSUN, Places365, Textures) as OOD. What happens if CIFAR 10 and CIFAR 100 — two similar datasets but differ in complexity — are used as ID and OOD? Would ASH methods still work in this case? We run ASH-S experiments with varying thresholds; results are seen in Figure 9. We can see that ASH still works in this rather difficult case. We are only including simple baselines like Energy and Softmax here.

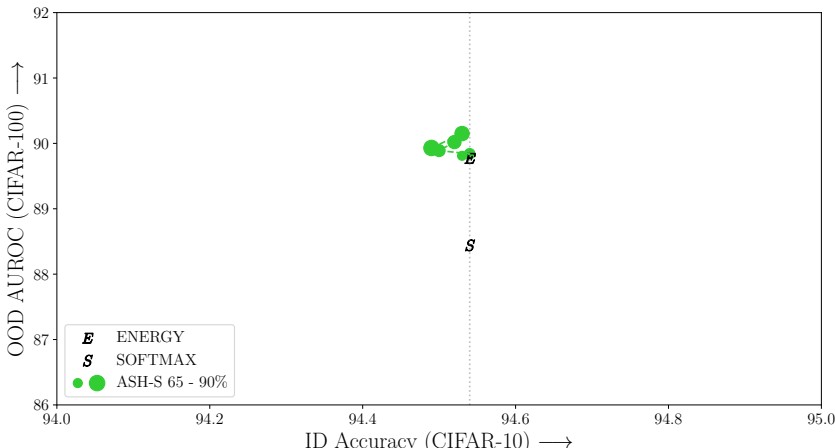

Figure 9: **ID-OOD tradeoff on DenseNet-101 architecture.** OOD detection rate (AUROC; on CIFAR-100) vs ID classification accuracy (Top-1 accuracy in percentage on CIFAR-10 validation set) of ASH-S variant used in this paper. Baseline methods "E" and "S" lie on the upper bound of ID accuracy (indicated by the dotted gray line) since it makes no modification of the network or the features. ASH-S (dots connected with dashed lines; smaller dots indicate lower pruning level) improves OOD detection and comes with slight drop in accuracy.

## J   ASH IMPROVES KNN FOR OOD

In Table 5 we showed that the ASH treatment improves upon existing methods: Softmax score, ODIN, energy score, and ReAct. In this section we experiment adding ASH to a methodologically different detector, the K Nearest Neighbor (KNN). Following the setup in (Sun et al., 2022), we add ASH to a ResNet18, trained on CIFAR-10. The OOD datasets are SVHN, LSUN, iSUN, Textures, Places365 (same as used in (Sun et al., 2022)). Since the original "penultimate layer of ResNet18" is now the final layer that generates feature vectors used for clustering, we placed ASH on the layer before that,

right after the 4th Block of the ResNet18. As we can see in Table 13, KNN with ASH treatments improves upon the baseline (KNN only) for a series of pruning percentages, more reliable for ASH-S than ASH-B.

| | OOD Datasets | | | | | | | | | | | |
|---|---|---|---|---|---|---|---|---|---|---|---|---|
| Method | SVHN | | LSUN | | iSUN | | Textures | | Places365 | | Average | |
| | FPR95 ↓ | AUROC ↑ | FPR95 ↓ | AUROC ↑ | FPR95 ↓ | AUROC ↑ | FPR95 ↓ | AUROC ↑ | FPR95 ↓ | AUROC ↑ | FPR95 ↓ | AUROC ↑ |
| Baseline (KNN only) | 27.97 | 95.48 | 18.50 | 96.84 | 24.68 | 95.52 | 26.74 | 94.96 | 44.56 | 90.85 | 28.49 | 94.73 |
| ASH-S@65 | 27.38 | 95.55 | 17.58 | 96.96 | 24.68 | 95.53 | 26.45 | 95.00 | 44.42 | 90.89 | 28.10 | 94.78 |
| ASH-S@70 | 26.71 | 95.62 | 16.80 | 97.06 | 24.48 | 95.53 | 26.26 | 95.00 | 43.99 | 90.96 | 27.65 | 94.84 |
| ASH-S@75 | 25.86 | 95.70 | 15.97 | 97.18 | 24.67 | 95.50 | 26.22 | 94.97 | 43.46 | 91.04 | 27.24 | 94.88 |
| ASH-S@80 | 25.06 | 95.79 | 14.84 | 97.31 | 25.02 | 95.38 | 26.21 | 94.88 | 43.35 | 91.10 | 26.90 | 94.89 |
| ASH-S@85 | 24.61 | 95.86 | 14.07 | 97.43 | 26.64 | 95.13 | 26.74 | 94.70 | 43.78 | 91.08 | 27.17 | 94.84 |
| ASH-S@90 | 24.26 | 95.93 | 13.49 | 97.53 | 30.11 | 94.59 | 28.09 | 94.34 | 44.97 | 90.87 | 28.18 | 94.65 |
| ASH-B@40 | 11.21 | 97.70 | 4.38 | 99.11 | 27.29 | 95.48 | 24.02 | 94.99 | 40.72 | 92.22 | 21.53 | 95.90 |
| ASH-B@65 | 37.21 | 93.65 | 33.75 | 93.40 | 24.67 | 95.83 | 27.71 | 94.40 | 52.25 | 89.31 | 35.12 | 93.32 |
| ASH-B@90 | 49.13 | 92.78 | 27.25 | 95.80 | 31.78 | 94.83 | 31.90 | 94.01 | 47.64 | 90.48 | 37.54 | 93.58 |

Table 13: **ASH improves upon KNN for OOD detction.** We add ASH treatments to the KNN algorithm for OOD detection, following the setup in (Sun et al., 2022). The network is a ResNet-18 pretrained on CIFAR-10. ASH-S with pruning percentages for 65 to 90 consistently show improvements. ASH-B with a lower percentage (40) greatly improves upon the benchmark, while larger percentages do not.

## K    DIFFERENCES BETWEEN ASH AND DICE

Here we highlight the key differences between ASH (our work) and DICE (Sun & Li, 2022), in both performance and methodology:

1. (Performance) ASH overall performs much better, and does not come with an ID accuracy degradation (Figure 2)

2. (Performance) DICE alone does not perform that well, but only when combined with ReAct, when doubles the overhead of hyperparameter tuning

3. (Methodology) ASH is completely on-the-fly, and does not require precomputation as DICE does

4. (Methodology) ASH make no modification of the trained network, with no necessary access to training data

We consider DICE a concurrent work as opposed to a precursor, which ASH comprehensively outperforms, albeit being philosophically similar.

## L    CALL FOR EXPLANATION AND VALIDATION

We are releasing two calls alongside this paper to encourage, increase, and broaden the reach of scientific interactions and collaborations. The two calls are an invitation for fellow researchers to address two questions that are not yet sufficiently answered by this work:

- What are plausible explanations of the effectiveness of ASH, a simple activation pruning and readjusting technique, on ID and OOD tasks?

- Are there other research domains, application areas, topics and tasks where ASH (or a similar procedure) is applicable, and what are the findings?

Answers to these calls will be carefully reviewed and selectively included in future versions of this paper, where individual contributors will be invited to collaborate. [4]. For each call we provide possible directions to explore the answer, however, we encourage novel quests beyond what's suggested below.

---

[4]Please follow instructions when submitting to the calls: `https://andrijazz.github.io/ash/#calls`

**Call for explanation**    A plausible explanation of the effectiveness of ASH is that the knowingly overparameterized neural networks likely overdo representation learning—generating features for data that are largely redundant for the optimization task at hand. It is both an advantage and a peril: on the one hand the representation is less likely to overfit to a single task and might retain more potential to generalize, but on the other hand it serves as a poorer discriminator between data seen and unseen.

**Call for validation in other domains**    We think adjacent domains that use a deep neural network (or any similar intelligent systems) to learn representations of data when optimizing for a training task would be fertile ground for validating ASH. A straightforward domain is natural language processing, where pretrained language models are often adapted for downstream tasks. Are native representations learned with those large language models simplifiable? Would reshaping of activations (in the case of transformer-based language models, activations can be keys, values or queries) enhance or damage the performance?

