# OpenReview forum: "Extremely Simple Activation Shaping for Out-of-Distribution Detection"
_ICLR.cc/2023/Conference — ICLR 2023 poster_

### Official Review · Reviewer_UzHM · 2022-10-24

**Confidence:** 4
**Correctness:** 3
**Technical Novelty And Significance:** 2
**Empirical Novelty And Significance:** 2
**Recommendation:** 5

**Clarity, Quality, Novelty And Reproducibility:**

The clarity and presentation of the paper are good. I have doubts regarding the novelty of the work. The algorithm proposed is closely related to a few previous works [1,2]. A proper justification for "Why it is performing better" is also missing. Adding reasoning would make the paper stronger.

[1] Sun, Y., Guo, C. and Li, Y., 2021. React: Out-of-distribution detection with rectified activations. In NeurIPS 21.

[2] Sun, Y. and Li, Y., 2022. Dice: Leveraging sparsification for out-of-distribution detection. In ECCV 22.


**Strength And Weaknesses:**

Strengths:

1. Well written and easy to follow.
2. Empirical analysis supports the effectiveness of the proposed algorithms.
3. Simple post-hoc idea based on activation shaping.

Weakness:

1. Although the empirical results are quite significant, the paper lacks a proper justification for why simply shaping the activations provides an improvement in OOD detection. Previous works [1] and [2] have already shown that pruning the activations provide an improvement in OOD detection. To make the paper stronger, the authors should provide a clear justification on how scaling the activations is further helping.
2. In Algorithms 2 & 3 (ASH-B, ASH-S), after pruning non-negative values are scaled to some constant value. However, proper reasoning on how this constant is chosen is missing. Can the constant be chosen as some arbitrary value and still get similar OOD performance?

3. It is not clear how the value of the hyper-parameter p is chosen. I shall suggest the authors to provide more detailed explanation on this.

4. It would be also interesting to see the performance on Hard OOD detection tasks such as CIFAR-100 vs CIFAR-10. Currently, the paper mainly shows results on OOD datasets that are significantly different from ID data and easier to detect.

Although not a major concern, the related works section can be improved by adding some recently proposed post-hoc works such as [3].

[1] Sun, Y., Guo, C. and Li, Y., 2021. React: Out-of-distribution detection with rectified activations. In NeurIPS 21.

[2] Sun, Y. and Li, Y., 2022. Dice: Leveraging sparsification for out-of-distribution detection. In ECCV 22.

[3] Sun, Y., Ming, Y., Zhu, X. and Li, Y.. Out-of-distribution Detection with Deep Nearest Neighbors. In ICML 22.

**Summary Of The Paper:**

This work presents a post-hoc activation shaping strategy for improved out-of-distribution (OOD) detection. The technique mainly consists of pruning the activation vector for a given sample in a chosen particular layer (favorably the penultimate layer). The paper proposes three algorithmic variants for the purpose of pruning and shaping the activations. The work is supported by empirical results on standard benchmark datasets (CIFAR-10/100 and ImageNet-1k) and detailed ablations.


**Summary Of The Review:**

This work proposes a simple-to-use algorithm to prune activations in a post-hoc fashion to improve OOD detection. The method is empirically supported through experimental results on standard benchmarks but lacks reasoning and motivation behind the working principle of the algorithm. Hence, I recommend borderline reject. However, I am willing to increase my score based on the rebuttal.

---

> ### Author Response · Authors · 2022-11-14
> **Thank you; new experiments and justifications**
>
> Dear reviewer, we really appreciate your time and effort giving suggestions to our paper. We are glad that you think our paper is well-written, empirical results strong and effective, idea simple and effective. Please see our responses to your weakness concerns below.
>
> > Although the empirical results are quite significant, the paper lacks a proper justification for why simply shaping the activations provides an improvement in OOD detection. Previous works [1] and [2] have already shown that pruning the activations provide an improvement in OOD detection. To make the paper stronger, the authors should provide a clear justification on how scaling the activations is further helping.
>
> Thanks for your raising the question on lacking justification. Other reviewers have raised the same concern, and we have since enriched our Appendix E, which is now named “**Intuition and explanation of ASH**.” In it, a few subsections — “ASH as post hoc regularization”, “Connection to a modified ReLU” and “How ASH changes the score distribution” — are all our efforts on providing the intuitive background and explanation of ASH.
>
> We updated the main text to **highlight the importance of justification**: “The strong results of ASH prompt us to ask: why does making radical changes to a feature map improve OOD? Why does pruning away a large portion of features not affect accuracy? Are representations redundant to start with? We discuss useful interpretations of ASH in Appendix Section E.”
>
> We agree that our work bears certain similarity to DICE and ReAct, however we do not agree that they have shown “pruning the activations provide an improvement in OOD detection”. In ReAct, the activations are “rectified”, never pruned, and there’s no knowledge of sparsification helping OOD being conveyed. In DICE, which is more a concurrent work to ours (officially published recently at ECCV 2022 in October 2022), weights are being pruned, but not activation.
>
> One idea that goes beyond our scope of this paper, but would be interesting to see, is if **scaling alone helps with methods like DICE**, which already performs sparsification, albeit on weights. We tried a few scaling functions, and observed varied effects. We want to hold on reporting them until we understand them further. If you have any suggestions along the line of DICE (weight pruning) + activation scaling, we’d love to hear your thoughts! We will report new results as soon as we get them.
>
> > In Algorithms 2 & 3 (ASH-B, ASH-S), after pruning non-negative values are scaled to some constant value. However, proper reasoning on how this constant is chosen is missing. Can the constant be chosen as some arbitrary value and still get similar OOD performance?
>
> Thank you for your questions. The choice of scaling functions for ASH-B and ASH-S is reflecting the intuition to preserve the activation mass before and after pruning. Hence, for ASH-B we simply set unpruned values to a constant so that the sum remains unchanged. For ASH-S, we first started with the same idea, using a linear function f=s1/s2, to preserve the activation mass before and after pruning. It works reasonably well — beating DICE and on par with ReAct on ResNet with Imagenet. To bootstrap this gain, we then experimented with superlinear functions and found out that exp(s1/s2) works the best. Given this evidence, we recommend linear or superlinear functions of (s1/s2) as a reliable choice with unseen applications. We have since added a section “Additional scaling functions” to explain our choice, as well as present results with other functions, thanks to your suggestion. As to arbitrary values, that's an interesting thought! Our experiments on the extreme variant ASH-RAND, reported in Section 4.4, suggests that it would work.
>
> > It would be also interesting to see the performance on Hard OOD detection tasks such as CIFAR-100 vs CIFAR-10. Currently, the paper mainly shows results on OOD datasets that are significantly different from ID data and easier to detect.
>
> Interesting suggestion! We took your suggestion and ran ASH-S with p values between 65% and 90% for **CIFAR10 vs CIFAR 100**, and reported the result in **Figure 9 in Appendix Section I**. We can see that ASH still works in this extremely difficult case. No other OOD methods in literature have reported results in a CIFAR 10 vs. CIFAR 100 setting, and therefore we are only including simple baselines like Energy and Softmax, which we outperform.
>
> > It is not clear how the value of the hyper-parameter p is chosen
>
> The hyper-parameter p is a design choice like in any neural network pruning algorithm. Throughout all our experiments we tested a wide range of p values, and observed rather stable results (see Figure 2). We observed that the range 65% to 90% generally works well, with ASH-S having a preference on high pruning thresholds (90%) and ASH-B on low thresholds (65%). For future adopters, we would highly suggest starting with these two combinations.
>
> Thank you!

---

> > ### Author Response · Authors · 2022-11-19
> > **update**
> >
> > > the related works section can be improved by adding some recently proposed post-hoc works such as [3].
> > >
> > > [3] Sun, Y., Ming, Y., Zhu, X. and Li, Y.. Out-of-distribution Detection with Deep Nearest Neighbors. In ICML 22.
> >
> > Thank you for making us aware of this great work. We ended up **implementing ASH on top of K-Nearest Neighbors (KNN)**, and reported that ASH improves on KNN consistently. The new result are in **Appendix J: ASH Improves KNN for OOD**. And the corresponding paper you mentioned is now cited there. Thank you!

---

> > > ### Comment · Reviewer_UzHM · 2022-12-01
> > > **Thanks for your response**
> > >
> > > Thanks for the rebuttal. Additional experiments and analysis improve the quality of the work. However, I am still concerned about the technical novelty of the work. DICE uses a measure of contribution (weights * activation) for sparsification. I understand that ASH prunes only the activations, but that is quite similar to the functioning of DICE. For the justification, I would have expected a more theoretical approach.
> > >
> > > Hence, I am not increasing my original score.

---

### Official Review · Reviewer_Pwvn · 2022-10-24

**Confidence:** 3
**Correctness:** 3
**Technical Novelty And Significance:** 2
**Empirical Novelty And Significance:** 4
**Recommendation:** 8

**Clarity, Quality, Novelty And Reproducibility:**

Clarity is good. The content is well-organized and is easy to follow. Quality and Reproducibility are good. This work provides extensive experimental evaluations and sufficient details. Novelty is fair.

**Strength And Weaknesses:**

Strength:

- The proposed method is simple and effective.
- The paper is well written.
- This work presents extensive empirical evaluations.

Weakness:

1. The choice of hyperparameter $p$ is mentioned in both the main text and the appendix. The authors did not discuss how the ID influences the choice of $p$. If only ID data is available, I think the hyperparameter selection scheme is important.
2. If the OOD detection task is CIFAR10 vs CIFAR100, does the ID-OOD trade-off still hold?
3. Table 5 shows that ASH improves on existing methods. Does ASH improve KNN? Does the choice of $p$ have a significant effect on the performance of KNN+ASH?

**Summary Of The Paper:**

 The authors hypothesize that over-parameterized DNNs extract too many features and can prune a large proportion of the features without drastically reducing ID accuracy, while significantly improving the OOD detection task. They propose a simple activation method to validate this hypothesis.  The proposed method prunes a large portion of a sample’s activation and rescaling the rest activation. The empirical evaluation on ImageNet shows that the improvement of the proposed methods is significant.

**Summary Of The Review:**

I recommend acceptance. The experimental evidence is strong.

---

> ### Author Response · Authors · 2022-11-14
> **Thank you; new CIFAR 10 vs 100 experiment**
>
> Dear reviewer, we thank you for considering our paper well written, proposed method simple and effective, and the evaluations extensive. We took your suggestions to heart and hope to engage with you further on how to further improve our paper.
>
> > The choice of hyperparameter p is mentioned in both the main text and the appendix. The authors did not discuss how the ID influences the choice of p. If only ID data is available, I think the hyperparameter selection scheme is important.
>
> Thank you for your question. We don't quite understand "the choice of hyperparameter p is mentioned in both the main text and the appendix" — did you mean that we had repeated texts describing the choice of p? We are sorry if we missed your point there.
>
> The answer to “how the ID influences the choice of p” is that **ID does not influence the choice of p at all**. We develop the method to be 100% post-hoc, meaning we only need access to the trained model, and nothing else (not ID or OOD data). We think as the scale of training goes up, **access to training data will become impossible**. We are already observing this from modern scale models like BERT, GPT, and DALL-E, none of which has disclosed their training data.
>
> Throughout the paper we consistently observed fair performances from ASH-S at a high pruning percentage, e.g. 90 and ASH-B at a moderate percentage, e.g. 65. And those choices seem to be invariant to model architecture, ID data, and OOD data. Hope that answers your question on hyperparameter selection scheme, which, again, is entirely ID data independent.
>
> > If the OOD detection task is CIFAR10 vs CIFAR100, does the ID-OOD trade-off still hold?
>
> Great idea! We took your suggestion and ran ASH-S with p values between 65% and 90% for CIFAR10 vs CIFAR 100, and reported the result in **Figure 9 in Appendix Section I**. We can see that **ASH still works** in this difficult case. No other OOD methods we are comparing to (DICE, ReAct, etc.) have reported results in a CIFAR 10 vs. CIFAR 100 setting, and therefore we are only including simple baselines like Energy and Softmax, which we outperform.
>
> > Does ASH improve KNN? Does the choice of p have a significant effect on the performance of KNN+ASH?
>
> That's an interesting suggestion. We are aware of one work "Out-of-Distribution Detection with Deep Nearest Neighbors" that uses KNN to tackle OOD detection. At this moment we are still actively trying to reproduce their results and adding ASH to it. We will report back as soon as we obtain any results. We agree that it will be great to know if this combo works well, or at all.
>
> [1] https://arxiv.org/abs/2204.06507

---

> > ### Comment · Reviewer_Pwvn · 2022-11-17
> > **Reply**
> >
> > > The answer to “how the ID influences the choice of p” is that ID does not influence the choice of p at all. We develop the method to be 100% post-hoc, meaning we only need access to the trained model, and nothing else (not ID or OOD data). We think as the scale of training goes up, access to training data will become impossible. We are already observing this from modern scale models like BERT, GPT, and DALL-E, none of which has disclosed their training data.
> >
> > **Ans:** Without access to validation data sampled from the ID distribution, how to determine the threshold value in the OOD detector? In general, the threshold value is the 5% quantile of the distribution of detection score under the ID distribution.

---

> > > ### Author Response · Authors · 2022-11-18
> > > **threshold value**
> > >
> > > > Without access to validation data sampled from the ID distribution, how to determine the threshold value in the OOD detector? In general, the threshold value is the 5% quantile of the distribution of detection score under the ID distribution.
> > >
> > > I see! We misunderstood your question. You are referring to the threshold on the score output by an OOD detector. In our case, that would be the energy score obtained with sparsified activations. Indeed, the current knowledge is to use 5% quantile of the distribution of detection score. In all of our experiments, we used threshold-free metrics, like AUROC, to evaluate our detector's performance, *taking account of all possible thresholds*. The hyperparameter p, in this case, is decided upfront, and *not tuned with ID data*. If we were to tune it, though, the algorithm will be similar to ReAct, where pre-computation with ID data is needed. That would be an interesting avenue to explore. Hope that answers your question.

---

> ### Author Response · Authors · 2022-11-19
> **Update: KNN + ASH results**
>
> Dear reviewer,
>
> Your good question **"Does ASH improve KNN?"** prompted us to start a series of experiments, and we can finally report back: yes, **ASH does improve upon KNN for CIFAR10**.
>
> We took the baseline method from "Out-of-Distribution Detection with Deep Nearest Neighbors" [1] and its corresponding codebase [2] to carry out our experiments. The ID dataset is CIFAR-10, and OOD datasets are SVHN, LSUN, iSUN, Textures, Places365 (same as used in [1]). Since the original penultimate layer of ResNet18 is now the final layer that generates feature vectors used for clustering, we placed ASH on the layer before that, right after the 4th Block of a ResNet18. The result we observed so far is very promising. As we can see in **Appendix J "ASH improves KNN for OOD"**, ASH+KNN improves upon the baseline (KNN only) for a series of pruning percentages, and for both ASH-S and ASH-B.
>
> However, we haven't be able to reach the same conclusion with ImageNet. As the time is limited, and each ImageNet experiment took 5-6 hours on a 1080Ti GPU, we could only afford to run a small number of experiments, and none of them so far is able to beat baseline. We will continue working on this experiment if time permits, and will spend time understanding the discrepancy between medium-scale and large-scale datasets/problems.
>
> [1] https://arxiv.org/abs/2204.06507
>
> [2] https://github.com/deeplearning-wisc/knn-ood

---

### Official Review · Reviewer_CKHN · 2022-10-24

**Confidence:** 5
**Correctness:** 4
**Technical Novelty And Significance:** 3
**Empirical Novelty And Significance:** 4
**Recommendation:** 8

**Clarity, Quality, Novelty And Reproducibility:**

About clarity and quality, this paper is clear to me, it is very well written and presented, and overall the presentation is very high quality.

About novelty, the novelty of this paper is the proposed ASH method for out of distribution detection in post-hoc way, producing state of the art performance in OOD detection with ImageNet and four OOD datasets (which are difficult), and also on CIFAR10/100 and six other OOD datasets (also difficult).

The proposed method is simple to understand and implement, and has very good performance. The only issue I see with the method is basically why it improves OOD detection performance, I believe the paper does not touch that subject, or introduce some intuition about this.

There is some similar previous work that the authors have references, like DICE which does weight sparsification, while the proposed method in this paper does activation sparsification, and it actually outperforms DICE. So I find that there is both a high degree of originality and novelty.

**Strength And Weaknesses:**

Strengths
- The paper is very well written, it is easy to understand, I have no more comments about writing.
- The proposed technique is very simple to understand and implement, prune activations with a threshold given by the p percentile, decide how to remove these activations, and then use the energy score to output a score for out of distribution detection.
- Post-hoc methods are preferable, as there is no need to retrain the model or use special training methods, and this method can be applied to any pre-trained model.
- The evaluation is solid, using multiple models, multiple ID (3) and multiple OOD datasets (10), multiple metrics (AUPR, AUROC, FP95), and a good selection of baselines, including many post-hoc methods. I do not have doubts about the validity of these results and conclusions. I think the only possible missing baselines are uncertainty-based method (Ensembles, Dropout, DUQ, etc), but I am not sure if these make sense given the post-hoc use of the proposed method.
- I really like Figure 2, it shows the trade-off between in-distribution accuracy and out of distribution performance, with state of the art methods and the proposed method at different activation pruning thresholds, clearly showing the advantage of the proposed method, but also showing how these methods compare by themselves. It is known that OOD detection methods might reduce performance but it is the first time I see such comparison. It is clear from this result that the proposed method has the best ID/OOD performance trade-off (best in both accuracy and AUROC).
- There is an improvement in out of distribution performance as measured by the AUROC and AUPR on ImageNet, in all benchmarks using ResNet, and in most benchmarks using MobileNets. For CIFAR10/100 using DenseNet, there is also a large improvement in all comparisons.
- There is a good set of ablation results to show the effect of some hyper-parameters, in particular the p percentile used as a threshold to shape/sparsify the activations, and also on which layer this method should be applied, which justifies the final decision made by the authors.

Weaknesses
After rebuttal, I  see no weaknesses.

~~- I think there is no theoretical justification provided in the paper on why this method works. By this I mean the paper does not do a theoretical analysis (which is fine), but the paper does not provide an intuition on why the method works. There is a good improvement in most out of distribution detection benchmarks, but then there is the question, why? Why does pruning or shaping activations over the p percentile improves OOD performance? Maybe the authors can give some intuition that guided the construction of the proposed method.~~

~~- I see that in Table 2, when using MobileNet, DICE outperforms your method in most benchmarks, I think this deserves some analysis or acknowledgment, it seems that the proposed method has some degree of dependency with the network architecture. Could the authors comment on this?~~

Minor Issues
- I think there is a double blind leak in page 18, in a footnote, I do not include this link here but this should not happen, I think the link is not anonymized and leaks some information about the authors.

~~- In Figure 2, please specify which OOD datasets were used to make this plot, the paper mentions four OOD datasets, but which ones?~~

~~- Table 5 shows combinations of the proposed method with other OOD detection methods (very nice), but why is there an improvement between energy score and energy score + ASH? The proposed method ASH already uses the energy score, this combination seems very strange to me. Please clarify.~~

**Summary Of The Paper:**

This paper is about weight pruning in a single layer and its relationship with out of distribution detection performance. The authors propose to prune activations at a single layer using the top p percentile, and candidate activations can either be set to zero (pruned), scaled, or binarized. Then the well known energy score is used to produce a score for out of distribution detection. This produces state of the art performance in out of distribution detection in all tested benchmarks.

The contributions of this paper are:
- A post-hoc and simple method for OOD detection based on activation shaping/sparsification.
- New state of the art results on three ID datasets and ten OOD datasets, including ImageNet, CIFAR10, and CIFAR100 as ID datasets over two different network architectures.
- A good evaluation and ablation results that justify some of the chosen hyperparameters and activation shaping
- A good study of the effect of activation shaping/sparsification and its relation to OOD performance. I think previous work only focuses on weight pruning/sparsification.


**Summary Of The Review:**

This is overall a good and high quality paper, the proposed method is a good improvement for out of distribution detection without requiring large modifications to the model or as training procedure. The proposed method is well studied, with many comparisons to the state of the art in a good selection of in-distribution and out-of-distribution datasets (including ImageNet and CIFAR10/100), and an excellent selection of ablations to justify the decisions made by the authors.

The only issues I find in the paper are minor, I mentioned them in weaknesses, mostly about intuition or analysis of why activation sparsification improves OOD performance, and some effects of network architecture on ASH performance.

After rebuttal, I can confirm that this paper should be accepted as the paper has improved.

---

> ### Author Response · Authors · 2022-11-12
> **Thank you; improved explanation**
>
> Dear reviewer, thank you for taking your time to give a very thorough and insightful review of our paper. We are very glad that you think paper is well written and presented strong results. As to the weaknesses and minor issues you pointed out, we took them very seriously, and have updated parts of the paper to improve it. Please see below for details.
>
> > [Weakness] ... the paper does not do a theoretical analysis (which is fine), but the paper does not provide an intuition on why the method works.
>
> Thank you for clarifying that theoretical analysis is not an absolute component in every work (“which is fine”). ASH is largely more empirical than theoretical, however, **the paper is not without intuitions and explanations on why our method works**. Due to space limit we moved the explanation section to Appendix E, where a few subsections, **“ASH as post hoc regularization”**, **“Connection to a modified ReLU”** and **“How ASH changes the score distribution”** are all our efforts on providing the intuitive background and explanation of ASH.
>
> Upon your suggestion, we renamed the section Appendix E from “Additional discussions” to “**Intuition and explanation of ASH**.” And added further emphasis in the main text, in the **end of Section 4.4**: “The strong results of ASH prompt us to ask: why does making radical changes to a feature map improve OOD? Why does pruning away a large portion of features not affect accuracy? Are representations redundant to start with? We discuss useful interpretations of ASH in Appendix Section E.”
>
> > [Weakness] ...in Table 2, when using MobileNet, DICE outperforms your method in most benchmarks, I think this deserves some analysis or acknowledgment
>
> Thank you for looking into Table 2 results carefully. On MobileNet, with 4 OOD datasets tested, DICE+ReAct wins on 2 (Sun, Places) and ASH wins the other 2 (INaturalist, Textures), and when averaged across all datasets, we agree that ASH falls slightly behind on OOD metrics (FPR95 and AUROC). However, what’s not shown in the table is that **ASH excels on ID accuracy metrics**, as Figure 2 indicates. In the caption of Table 2, we stated honestly that “For ResNet, ASH consistently performs better than benchmarks, across all the OOD datasets. In the case of MobileNet, ASH performs comparably with DICE+ReAct.”
>
> Upon your suggestion, we’ve **added in the main text acknowledging that DICE+ReAct performs slightly better** on averaged OOD metrics with MobileNet, and pointed out ASH’s strength over DICE+ReAct: algorithmically simpler, much lighter turning effort, zero precomputing cost, and preserving ID accuracy.
>
> > [Minor] I think there is a double blind leak in page 18
>
> **This is not a double blind leak.** The link on page 18 is referring to the source code of DICE, which was used when we implemented their method on additional architectures like MobileNet. It does not contain any information about the authors of this paper. We thank you so much though for carefully checking on this.
>
> > [Minor] In Figure 2, please specify which OOD datasets were used to make this plot
>
> We apologize for the unclear explanation. As described in Table 1, OOD datasets used to make this plot are: iNaturalist, SUN, Places365 and Textures. We have since **updated Figure 2 caption to clarify**. Thanks for your suggestion.
>
> > [Minor] Table 5 shows combinations of the proposed method with other OOD detection methods (very nice), but why is there an improvement between energy score and energy score + ASH? The proposed method ASH already uses the energy score, this combination seems very strange to me. Please clarify.
>
> Thank you for your question; we apologize for the lack of clarity. In Table 5, “Energy Score” means energy score alone, without any ASH treatments. “Energy Score + ASH” is the same as what we refer to as “ASH” everywhere else. Upon your suggestion, we **made two changes** to clarify: **1) changing the naming convention** from {score function} + ASH to {score function} + **ASH tr.**, where ASH tr. means the specific ASH treatment, aka pruning and scaling (explained in the caption); and 2) **adding in the table caption** “Note that "Energy score + ASH tr." is the same as what’s referred to as "ASH" elsewhere.”
>
> We hope the new changes have remedied the confusions and lack of clarification. We look forward to engaging with you further as to how to improve our paper. Thank you!

---

### Official Review · Reviewer_kqRP · 2022-10-25

**Confidence:** 4
**Correctness:** 3
**Technical Novelty And Significance:** 3
**Empirical Novelty And Significance:** 3
**Recommendation:** 3

**Clarity, Quality, Novelty And Reproducibility:**

This paper has a clear presentation, and the reproducibility is good.

The quality and novelty are not well supported by the experiment. The proposed method looks heuristic, meaning that motivation and evaluation should be stengthened.

**Strength And Weaknesses:**

Pros:

1. The paper is well-written.

2. The method is simple and easy to reproduce.

3. The experiments are sufficient and performance are quite well on both imagenet and cifar benchmark.


Cons:

1. In algorithm ASH-S, I think the scaling factor exp(s1/s2) is a hyperparameter. Are there other functions that can replace the exponential function? In other words, when the application scene changes, do the authors need to replace the exponential function to adapt to the current scene to obtain the optimum?

2. The paper designs three algorithm and mostly ASH-S performs best (like in Table 1 and Table 2). Sometimes, ASH-B performs better than others, like Table 1 (MobileNet). So I wonder how to choose the algorithm when meeting a new dataset and a new model?

3. As shown in Table 1, when the model is ResNet, ASH-S can achieve 95.12% AUROC and 22.18% FPR95. But when the model is MobileNet, the method can not outperform the SOTA. I think the generalization of these algorithms is not very good, and when the model changes, the performance difference is too large.

4. The placement in Figure 5 and the results in Table 6 (Appendix A) show that the method can not work when it is deployed in 1-3th Block of ResNet. The result in 4 Block is not good enough and the highest performance is in the penultimate Layer. I think the results show that the method only works in the penultimate Layer, and the generalization of insertions in other positions in backbone is extremely poor.

5. The placement of the method is in the penultimate Layer, which is the same with DICE. The method discards some features at the penultimate layer, which is actually partially equivalent to dropping out some connection weights at the penultimate layer (DICE's approach). So when this method is deployed to the penultimate layer, I think it has a high similarity to DICE.

6. The theoretical proof of why the algorithm works is insufficient, as the operations of Algorithms 2 and 3 seem to be the result of empirical design rather than rigorous theoretical derivation. The authors may consider improving the interpretability and theoretical basis of this article.


**Summary Of The Paper:**

The paper focuses on detecting OOD samples in the inference time by pruning a large portion of an input sample's activation and lightly adjusting the remaining. The method can be easily combined with previous OOD scores (like MSP, ODIN, Energy and ReAct). When combined with the energy score, it shows a significant improvement in OOD detection, on both moderate and large-scale image classification benchmarks.

**Summary Of The Review:**

The paper focuses on detecting OOD samples in the inference time by pruning a large portion of an input sample's activation and lightly adjusting the remaining. The method can be easily combined with previous OOD scores (like MSP, ODIN, Energy and ReAct). When combined with the energy score, it shows a significant improvement in OOD detection, on both moderate and large-scale image classification benchmarks. However, the motivation of the proposed method is not clear and not convincing.

---

> ### Author Response · Authors · 2022-11-11
> **Thank you; new experiments added [1/2]**
>
> Dear reviewer,
>
> Thank you for giving your precious time and energy reviewing our paper. We are glad that you found the paper well-written, idea simple and easy to reproduce, and experiments sufficient and our method's performance strong.
>
> To address some of your concerns, we have since conducted **additional experiments** and updated our paper. Please see below for our point-to-point rebuttal.
>
> > "1. In algorithm ASH-S, I think the scaling factor exp(s1/s2) is a hyperparameter. Are there other functions that can replace the exponential function? In other words, when the application scene changes, do the authors need to replace the exponential function to adapt to the current scene to obtain the optimum?"
>
> Thank you for your question. The scaling function is indeed one of the design choices the algorithm designer has to make. In our experiments, we observed that the scaling function in use, f=exp(s1/s2), is quite stable throughout all experiments, across datasets (CIFAR, ImageNet) and architectures (ResNet, MobileNet, DenseNet). The story of converging to this function is that we first started with the most intuitive, linear function f=s1/s2, to preserve the activation mass before and after pruning. It works reasonably well — beating DICE and on par with ReAct on ResNet with Imagenet. To bootstrap this gain, we then experimented with superlinear functions and found out that exp(s1/s2) works the best. Given this evidence, we recommend linear or superlinear functions of (s1/s2) as a reliable choice with unseen applications. **We have since added a section in Appendix H, “Additional scaling functions” to include ASH-S results with other scaling functions, thanks to your suggestion.**
>
> > "2. The paper designs three algorithm and mostly ASH-S performs best (like in Table 1 and Table 2). Sometimes, ASH-B performs better than others, like Table 1 (MobileNet). So I wonder how to choose the algorithm when meeting a new dataset and a new model?"
>
> Thank you for the keen observation in the performance variance between the two ASH algorithms. The conclusion from our extensive experiments is that these two variants **work comparable and are both preferable**: ASH-S at a high pruning percentage (90%) and ASH-B at a moderate percentage (65%). There’s not a clear winner between these two choices in terms of performance, as they perform similarly and both surpass other contemporary methods. However, they do come with **different tradeoffs**: one with a much larger pruning capacity, and the other makes use of binary representations. We think it is valuable to provide flexibility to future task designers, in that some tasks and implementations might favor a high level of sparsity, while others favor reduced bits of representation. Hence we opted to present both variants across all of our experiments, and showed evidence that they both perform reliably well.
>
> > "3. As shown in Table 1, when the model is ResNet, ASH-S can achieve 95.12% AUROC and 22.18% FPR95. But when the model is MobileNet, the method can not outperform the SOTA. I think the generalization of these algorithms is not very good, and when the model changes, the performance difference is too large."
>
> Thank you for the careful analysis of our ImageNet results. We agree that on ResNet, ASH is a more clear outperformer, but we have to respectfully disagree that the performance degrades for MobileNet and hence poses a generalization concern. On MobileNet, when we test on 4 OOD datasets, DICE+ReAct wins on 2 (Sun, Places) and ASH wins the other 2 (INaturalist, Textures), and even though the “average” numbers fall slightly behind, in the grand scheme we claim that ASH performs comparably with DICE+ReAct, aka “sharing the SOTA”. Furthermore, SOTA goes beyond comparing AUROC and FPR95. As we noted in Figure 2, **ASH does not destroy ID accuracy like all other methods**. Additionally, ASH presents a simpler algorithm with much lighter turning effort while DICE and ReAct each come with their own hyperparameters, precomputing cost, and the DICE+ReAct variant comes with an increased complexity (see implementation notes in Appendix F).
>
> To further validate the generalization ability of ASH, we have since added two more architectures to our ImageNet experiment upon your suggestion. **In Appendix G, we added results for VGG16 and DenseNet121**, showing that the same ASH algorithms without any further tuning of hyperparameters, produce reliable performance when architectures drastically change. Overall, for ImageNet, across 4 datasets (more than any contemporary paper) we show strong and consistent performance of ASH.
>
> [To be continued]

---

> > ### Author Response · Authors · 2022-11-11
> > **Thank you; new experiments added [2/2]**
> >
> > > "4. The placement in Figure 5 and the results in Table 6 (Appendix A) show that the method can not work when it is deployed in 1-3th Block of ResNet. The result in 4 Block is not good enough and the highest performance is in the penultimate Layer. I think the results show that the method only works in the penultimate Layer, and the generalization of insertions in other positions in backbone is extremely poor."
> >
> > We fully agree that ASH does not seem to work when placed in Block 1-3. The purpose of extensive experiments on the placement of ASH is to **show that the placement matters, and how**. We are sorry for the likely confusing message of indicating that ASH works everywhere, or of its generalization on locations. However, **ASH at Block 4 works reasonably well. We have since added a new row to Table 6, indicating that ASH-S and ASH-B have a comparable performance when placed in Block 4.** This seems to suggest that the later placement, the better, which makes intuitive sense as ASH works by simplifying image representations, and representations in later layers tend to be more stabilized and robust to perturbations.
> >
> >
> > > "5. The placement of the method is in the penultimate Layer, which is the same with DICE. The method discards some features at the penultimate layer, which is actually partially equivalent to dropping out some connection weights at the penultimate layer (DICE's approach). So when this method is deployed to the penultimate layer, I think it has a high similarity to DICE."
> >
> > We agree that our method has a certain similarity to DICE. **DICE is a concurrent work to ASH**, as it is only officially published recently at ECCV 2022 (October 2022). The true story is that we were towards the final development of ASH when we found out about DICE. We had since profusely studied, cited, included, and compared with DICE into our experiments, including reimplementing it on additional architectures for ImageNet (DICE paper only has only ResNet on ImageNet). That said, we would still like to highlight the key differences between ASH and DICE:
> >  - ASH overall performs much better, and does not come with an ID accuracy degradation (Figure 2)
> >  - ASH is completely on-the-fly, and does not require precomputation as DICE does
> >  - DICE alone does not perform that well, but only when combined with ReAct, when doubles the overhead of hyperparameter tuning
> >  - ASH make no modification of the trained network, with no necessary access to training data
> >
> > In sum, we consider DICE a concurrent work as opposed to a precursor, which ASH comprehensively outperforms, albeit being philosophically similar.
> >
> > > "6. The theoretical proof of why the algorithm works is insufficient, as the operations of Algorithms 2 and 3 seem to be the result of empirical design rather than rigorous theoretical derivation. The authors may consider improving the interpretability and theoretical basis of this article."
> >
> > We agree that ASH is largely more empirical work than theoretical. We derived ASH upon observing how ID and OOD data behave differently in the activation space, inspired by ReAct. “Why the algorithm works” is a prevailing question in all of deep learning research. In the paper we provided some hypotheses as to why pruning and simplifying activation works, as seen in Appendix E, **“ASH as post hoc regularization”**, **“Connection to a modified ReLU”** and **“How ASH changes the score distribution”**. We have since strengthened those arguments and made clearer pointers to them from the main body of the paper, thanks to your suggestion.
> >
> > Overall, we thank you for giving a thoughtful review of our paper, we have since added numerous **new experiments** to address your concern on the generalization of **scaling functions, architecture, and ASH placements**.
> >
> > We hope to engage with you further on other possible ways to improve our work. Thank you!

---

> > > ### Comment · Reviewer_kqRP · 2022-11-30
> > > **Further Comments**
> > >
> > > Thanks for your responses. Please see my follow-up comments below.
> > >
> > > Re A1: Firstly there are more choices of scaling functions than just linear and exponential functions, and the authors could have tried more functions with similar properties. Secondly, the authors did not try other scaling functions on other models and other datasets before concluding directly that "the scaling function in use, f=exp(s1/s2), is quite stable throughout all experiments, across datasets (CIFAR, ImageNet) and architectures (ResNet, MobileNet, DenseNet)". This is not sufficient to show that linear scaling functions are worse performing and more stable than exponential scaling functions across datasets (CIFAR, ImageNet) and architectures (ResNet, MobileNet, DenseNet).
> > >
> > > Re A2: The performance difference between ASH-B and ASH-S on different datasets or different models is still quite large. They just don't have much performance difference under the setting of ImageNet+ResNet50. But on ImageNet+MobileNet setting,  ASH-B has ~2-3% higher performance than ASH-S. Similarly, on Cifar benchmars, ASH-S is ~2-3% higher than ASH-B again. The authors did not address my doubts about how to choose between these two methods on new datasets or new models.
> > >
> > > Re A3: "On MobileNet, when we test on 4 OOD datasets, DICE+ReAct wins on 2 (Sun, Places) and ASH wins the other 2 (INaturalist, Textures)": In fact, ASH-B can only surpass SOTA on iNaturalist, and ASH-S can only surpass SOTA on Texture. Each individual method can only surpass SOTA on one dataset, which I don't think can be called "ASH wins two datasets". Looking at the difference in averages, the smallest difference on FPR95 between ASH and SOTA is ~4%, and I don't think this difference can be called " ASH performs comparably with DICE+ReAct”. Moreover, the experiments in Appendix G only use Energy as the baseline. Maybe adding DICE, React, DICE +React as baselines are better.
> > >
> > > Re A4: The placement of ASH is critical for the performance. The result in 4 Block is not good enough and doesn't outperform SOTA. So I think ASH can only be placed at the penultimate layer.
> > >
> > > Re A5: DICE is publicly available online in November 2021, and the core operations of DICE and this article are still too similar. The method discards some features at the penultimate layer, which is actually partially equivalent to dropping out some connection weights at the penultimate layer (DICE's approach). Thus, I think the contribution of this article is incremental.
> > >
> > > In general, **the novelty of this paper is concerned due to DICE**. It is hard to consider this work as a co-current work of DICE that is available online in Nov. 2021. The method discards some features at the penultimate layer, which is actually partially equivalent to dropping out some connection weights at the penultimate layer (DICE's approach).

---

### Author Response · Authors · 2022-11-18
**Summary of added experiments and updates**

We want to thank the reviewers for their careful read of our work and valuable suggestions. After consideration of their feedback, we have added a great number of new experiments and results which we believe have greatly strengthened our overall contribution.

 - **Additional scaling functions (Appendix H)**:  we include ASH-S results with other scaling functions like linear, thanks to the suggestions of *Reviewer kqRP*.
 - **Additional ImageNet architectures (Appendix G)**: we added results for VGG16 and DenseNet121, on ImageNet, proving that ASH does further generalize across architectures, thanks to the suggestions of *Reviewer kqRP*.
 - **New figure on ID-OOD tradeoff (Appendix C)**: we added the equivalent of Figure 2 for MobileNet, indicating that again ASH not only performs well on OOD detection metrics, but does not deteriorate ID performance.
 - **Intuition and explanation of ASH (Appendix E)**: since many reviewers (*Reviewer kqRP, CKHN, UzHM*) pointed out that the paper is weak on the justification and intuition of ASH, we have highlighted a number of justifications in this section.
 - **New ASH placement (new row to Table 6)**, indicating that ASH-S and ASH-B have a reasonable performance when placed in Block 4, in answer to *Reviewer kqRP*.
 - **New CIFAR10 vs CIFAR 100 experiment (Appendix I)**: we have added the CIFAR 10 vs CIFAR 100 OOD task, and shown that ASH works reasonably well, thanks to the suggestions of *Reviewer Pwvn* and *UzHM*.
 - **New ASH + KNN results (Appendix J, Table 13)**: thanks to the suggestion of *Reviewer Pwvn*, we carried out experiments to see if ASH provides improvements on top of a K Nearest Neighbor (KNN) algorithm for OOD detection. We are able to reproduce results from an existing paper, added ASH treatment to it, and concluded with results that ASH indeed provide an immediate boost compared to the baseline KNN in the CIFAR-10 setting.

We also made corrections and clarifications throughout the paper per review feedback. Thank you.

---

### Decision · Program_Chairs · 2023-01-20

**Decision:**

Accept: poster

**Justification For Why Not Higher Score:**

Some concerns around novelty of approach.

**Justification For Why Not Lower Score:**

Results are strong and worthwhile for communication to the community.

**Metareview: Summary, Strengths And Weaknesses:**

Paper Summary:
Prior art has proposed DICE, a method that imposes sparsity on penultimate layers of neural networks, to enhance OOD discrimination. The prior art learns this pattern on a subset of data and then fixes the sparse signals.

The current work proposes a new variation of this idea, except sparsity is introduced on-the-fly, dynamically on an instance-by-instance level. Significant improvements in performance are demonstrated over the prior state-of-art.


Review Summary:

Pros:
- Well written (kqRP, CKHN, Pwvn, UzHM)
- Experiments simple and effective, easy to reproduce (kqRP, CKHN, Pwvn, UzHM)
- Experiments are sufficient and performance improvement is good (kqRP, CKHN, Pwvn, UzHM)
- Method can be combined with previous OOD approaches (kqRP)
- Good study on the effects of activation shaping, which was lacking in prior works (CKHN)
- Post-hoc methods such as this are preferable to other methods (CKHN)
- Fig 2 is very good and shows the clear benefit (CKHN)
- Good work on ablation experiments to show the effects of hyperparameters (CKHN)

Cons:
- Scaling factor may be a hyperparameter. Further details on why this was chosen and other hyperparameters would be beneficial (kqRP, Pwvn, UzHM) -- Authors responded with more details about other approaches studied and included more information in the appendix.
- Not clear which sub-method works best or should be chosen for application (kqRP). -- Authors responded that two submethods were clearly winners, but which performed best was indeed a design choice. AC wonders if an ensemble of these two approaches would yield even better results.
- Results may not generalize across architectures (kqRP) -- Authors clarified that with MobileNet architecture, while the performance is comparable to SOTA (not better), the proposed approach still has other benefits, such as eliminating the need for any training, and maintaining better ID performance. Authors added more architectures and showed that overall, there is clear benefit.
- Method only works in penultimate layer, therefore method is same as DICE since it only works in penultimate later (kqRP, UzHM) -- Authors have better clarified distinctions with DICE. In AC opinion, the most clear distinction is that ASH runs "on the fly" and requires no additional training or model modifications as DICE, and yields significantly improved results. While there are similarities in the fundamental approaches, AC recognizes that small changes are not always intuitive, and in this case, the impact on the ease of implementation and performance is significant and noteworthy.
- Theoretical explanations are insufficient. Paper presents mostly empirical evidence (kqRP, UzHM) -- Authors have expanded this discussion in the appendix.

AC Recommendation: While reviews are mixed, in AC opinion, this is a clear accept, in line with reviewers CKHN and Pwvn. Reviewers (kqRP,UzHM) concerns about novelty as primary reason for rejection are noted and appreciated. On a fundamental level this technique shares ideas with prior work DICE. However, the changes in this work are still unique and impactful not only on the model's improved overall performance, but in the way it eliminates need of prior training. In summary paper is noteworthy and useful to the community. AC requests authors to make changes to paper to more clearly recognize DICE and the differences between the two methods (perhaps a table).

**Note From Pc:**

if the above contains the word "oral" or "spotlight" please see: "oral" presentation means -> notable-top-5% and "spotlight" means -> notable-top-25%. As stated in our emails, we are disassociating presentation type from AC recommendations

---

> ### Author Response · Authors · 2023-03-01
> **Requested change made in camera ready**
>
> Dear Program Chairs,
>
> Thanks for the decision to accept our paper. We want to let you know that in our camera ready version, we added Section K in Appendix, "Differences between ASH and DICE" to answer your request "AC requests authors to make changes to paper to more clearly recognize DICE and the differences between the two methods (perhaps a table)."
>
> Thank you again for helping improve this work.